# Synaptic deregulation of cholinergic projection neurons causes olfactory dysfunction across five fly Parkinsonism models

Ulrike Pech[1,2†], Jasper Janssens[1,3,4†], Nils Schoovaerts[1,2], Sabine Kuenen[1,2], Carles Calatayud Aristoy[1,2], Sandra F Gallego[1,2], Samira Makhzami[1,3,4], Gert J Hulselmans[1,3,4], Suresh Poovathingal[1,3,4,5], Kristofer Davie[1,3,4,5], Adekunle T Bademosi[1,2], Jef Swerts[1,2], Sven Vilain[1,2], Stein Aerts[1,3,4], Patrik Verstreken[1,2]*

[1]VIB-KU Leuven Center for Brain and Disease Research, Leuven, Belgium; [2]KU Leuven, Department of Neurosciences, Leuven Brain Institute, Leuven, Belgium; [3]KU Leuven, Department of Human Genetics, Leuven Brain Institute, Leuven, Belgium; [4]VIB-KU Leuven Center for AI and Computational Biology (VIB.AI), Leuven, Belgium; [5]VIB-KU Leuven Center for Brain and Disease Research Technologies, Single Cell, Microfluidics and Bioinformatics Expertise Units, Leuven, Belgium

*For correspondence: patrik.verstreken@kuleuven.be

†These authors contributed equally to this work

## eLife Assessment

This is an **important** study demonstrating that anosmia in Parkinson's disease patients is due to dysfunction in cholinergic neurons. This study provides **compelling** evidence, using scRNA sequencing, that cholinergic olfactory projection neurons (OPN) are consistently affected in five different fruit fly models of Parkinson's disease, exhibiting synaptic dysfunction before the onset of motor deficits. Comparisons with scRNA sequencing of patients' human brain samples reveals similar synaptic gene deregulation in cholinergic neurons of patients. This study points the possibility that targeting cholinergic neurons could be a potential avenue for early diagnosis and intervention in PD.

**Abstract** The classical diagnosis of Parkinsonism is based on motor symptoms that are the consequence of nigrostriatal pathway dysfunction and reduced dopaminergic output. However, a decade prior to the emergence of motor issues, patients frequently experience non-motor symptoms, such as a reduced sense of smell (hyposmia). The cellular and molecular bases for these early defects remain enigmatic. To explore this, we developed a new collection of five fruit fly models of familial Parkinsonism and conducted single-cell RNA sequencing on young brains of these models. Interestingly, cholinergic projection neurons are the most vulnerable cells, and genes associated with presynaptic function are the most deregulated. Additional single nucleus sequencing of three specific brain regions of Parkinson's disease patients confirms these findings. Indeed, the disturbances lead to early synaptic dysfunction, notably affecting cholinergic olfactory projection neurons crucial for olfactory function in flies. Correcting these defects specifically in olfactory cholinergic interneurons in flies or inducing cholinergic signaling in Parkinson mutant human induced dopaminergic neurons in vitro using nicotine, both rescue age-dependent dopaminergic neuron decline. Hence, our research uncovers that one of the earliest indicators of disease in five different models of familial Parkinsonism is synaptic dysfunction in higher-order cholinergic projection neurons and this

contributes to the development of hyposmia. Furthermore, the shared pathways of synaptic failure in these cholinergic neurons ultimately contribute to dopaminergic dysfunction later in life.

## Introduction

Parkinson's disease (PD) is a complex multifactorial neurodegenerative disease that affects millions. Dopaminergic neuron (DAN) death in the *Substantia nigra* causes motor defects in patients. However, 97% of patients suffer from non-motoric dysfunction decades earlier, including constipation, REM sleep disorder, and hyposmia (*Ansari and Johnson, 1975*; *Chase and Markopoulou, 2020*; *Doty et al., 1988*; *Parkinson, 2002*; *Schapira et al., 2017*). These non-motor defects in PD are often overlooked because they occur before disease diagnosis and, thus, they do not belong to the normal clinical work-up. Furthermore, non-motoric defects are largely nonresponsive to dopamine replacement therapy suggesting other cell types are involved (*Kalia and Lang, 2015*).

Hyposmia, the reduced ability to discern odors, is one of the earliest defects associated with familial and sporadic Parkinsonism (*Chase and Markopoulou, 2020*; *Doty, 2012*; *Doty et al., 1988*; *Haehner et al., 2009*). These patients show difficulties to classify odors: they often perceive banana odor as motor oil or beer odor as fruit in standardized odor identification tests (UPSIT) (*Double et al., 2003*). These observations suggest that hyposmia in Parkinsonism is caused by a defect in higher-order perception, consistent with the observations that first-order olfactory receptor cells of the nasal epithelium are not majorly affected in patients (*Mueller et al., 2005*; *Witt et al., 2009*). The synaptic site of second-order olfactory neurons, the olfactory bulb, is modulated by basal forebrain cholinergic neurons and local GABAergic and dopaminergic interneurons (*Harvey and Heinbockel, 2018*). While these DAN do not degenerate in Parkinsonism, it has been suggested there is local impairment of cholinergic transmission in the olfactory bulb of Parkinsonism patients and vertebrate models (*Huisman et al., 2004*; *Mundiñano et al., 2013*; *Zhang et al., 2015*), but the molecular defects are elusive.

There are only few models to study hyposmia in Parkinsonism and most are based on the use of toxins or expression of mutant α-Synuclein (*Taguchi et al., 2020*). While this may recapitulate aspects of the disease, they bias the disease mechanisms to known phenomena associated with the specific model. Modeling approaches to Parkinsonism are further hindered due to the large number of mutated genes. These mutations affect various biological functions and therefore it is currently hard to unveil the specific and possibly common pathways and cell types that are at the basis of the early defects in the disease. So far, mutations in more than 25 genes have been identified that cause familial forms of Parkinsonism (*Brooker et al., 2024*). These genes cause defects in a diverse range of processes including mitochondrial function, endocytosis, lysosomal function, autophagy, and protein translation (*Kalia and Lang, 2015*). Nonetheless, these mutations do result in overlapping motor and non-motor defects, including hyposmia. This raises the question of whether defective mechanisms across the genetic space of Parkinsonism converge on common biological processes and affect similar cell types beyond DAN. While we are in the process of generating a comprehensive collection of fruit flies with PD knock-out and knock-in mutations (see also *Kaempf et al., 2024*), this study focuses on the first five *Drosophila* Parkinsonism models. Using these models, we investigated cellular and molecular dysfunctions that precede motor defects and find that early cholinergic projection neuron problems contribute to dopaminergic system failure later in life. The mutants included in this study are 'classical' Parkinson disease genes, like *LRRK2* and *PINK1*, and Parkinsonism genes that affect vesicle trafficking processes, including *SYNJ1*, *DNAJC6/Aux.* and *RAB39B*. Additionally, we also analyzed postmortem human brain samples from idiopathic patients with *LRRK2* variants and iPSC-derived human DAN with a pathogenic *LRRK2* mutation.

## Results

### A new collection of five familial PD knock-in models

The characterization of cells and pathways affected in PD is hindered because patient populations are heterogeneous and because mutations in numerous genes have been linked to familial forms of the disease (*Brooker et al., 2024*). To overcome these problems, we created a new collection of *Drosophila* PD knock-in models carrying pathogenic mutations using a genetic standardized and

controlled design, enabling direct side-by-side comparative analyses. Additionally, these were extensively backcrossed into one isogenic background (>10 generations) and did not rely on overexpression of proteins (e.g. α-Synuclein). We selected homologues of five PD genes and replaced the *Drosophila lrrk, rab39, auxilin (aux), synaptojanin (synj)*, and *pink1* genes by wild-type and pathogenic mutant human or *Drosophila* coding DNA (cDNA) sequences (CDS) at the endogenous locus (*Figure 1a*; *wild-type hPINK1* was a UAS construct; see Materials and methods). We confirm that the wild-type and pathogenic mutant human CDS (*LRRK2, RAB39B*, and *PINK1*) or *Drosophila* CDS (*DNAJC6 (Aux)* and *SYNJ1 (Synj)*) are expressed at endogenous levels (*Figure 1—figure supplement 1a*, *Supplementary file 1*). To determine if our new models recapitulate known PD-relevant phenotypes, we tested them in an array of assays that were previously used to analyze other fly PD models. Wild-type knock-ins are similar to controls (background fly line), indicating the cDNA knock-ins (human or fly) recapitulate normal gene function. Conversely, the PD knock-in models show defects akin to those seen in previously described mutants of these genes. For example, we find disturbances in mitochondrial membrane potential and electrophysiological defects in electroretinogram (ERG) recordings of flies stressed by exposure to constant light for 7 days (*Hindle et al., 2013*; *Morais et al., 2009*; *Mortiboys et al., 2010*; *Ng et al., 2012*; *Vanhauwaert et al., 2017*; *Figure 1b*, *Figure 1—figure supplement 1b–d*). Note that this specific experimental ERG setup differs from studies involving the progressive aging of PD mutant flies (*Jacquemyn et al., 2023*). Importantly, we also tested the flies for their performance in a startle-induced negative geotaxis (SING) assay. This is a motor behavior that depends on central brain DAN (**p**osterior-**a**nterior-**m**edial DAN: 'PAM'). Previous work showed that PAM-dependent SING was unaffected in young flies that express α-Synuclein, but that SING declines as these flies age (*Riemensperger et al., 2013*). Similarly, none of our models shows a SING defect at a young age (5 days, young) (*Figure 1b*). However, as flies get older (25 days, old), the PD knock-ins display impaired performance in this DAN-dependent locomotion assay, while the performance of controls is not affected (*Figure 1b*). This is in line with recent work where we show that SING defects in PD mutant models are rescued when the flies are fed L-DOPA but not D-DOPA, indicating a very strong correlation between SING defects and defects in dopaminergic synaptic innervation of PAM DAN onto mushroom body neurons (*Kaempf et al., 2024*). Taken together, our data suggests that the mutants we used in this study suffer from a progressive locomotion defect that is linked to DAN synapse dysfunction.

## Cholinergic olfactory projection neurons are impaired across five PD mutant fly models

To define cell types affected by pathogenic PD mutations early in life, we used an unbiased approach and conducted single-cell RNA sequencing (RNA-seq) of entire brains of young (5 days) animals (*Supplementary file 2*). We pooled 108k newly sequenced cells from these five mutants and controls with the 118k wild-type cells from our original atlas leading to 186 cell clusters, of which 81 were annotated to known cell types (*Davie et al., 2018*; *Janssens et al., 2022*; *Figure 2a*). PD mutant cells equally mix with the cells of the original fly brain cell atlas, and the frequency of recovered cell types is similar between control and mutant brains (including DAN; *Figure 2b–f*, *Figure 2—figure supplement 1a and b*; *Supplementary file 3*). We further do not find any cell type that shows higher or lower expression of the knocked-in CDS (*Figure 2—figure supplement 1c*). Thus, cellular identity and cellular composition are preserved in young PD fly models.

For every cluster, we then compared gene expression between cells from mutants and controls using two independent algorithms (Wilcoxon and DESEQ2, *Wang et al., 2019*) and find both methods correlate well (average Spearman correlation of signed p-value>0.8 to >0.95; *Figure 2—figure supplement 1d*). However, the number of differentially expressed genes (DEGs) is largely dependent on the number of cells present in a cluster. To correct this bias, we fitted a negative binomial model that correlated the number of DEG and cell cluster size in each mutant (*Figure 2—figure supplement 1e*, *Supplementary file 4*). Using the residuals of this model we find eight common predominantly affected cell types across the five mutants (*Supplementary file 5* and *Supplementary file 6*). We also find cell types uniquely affected in some mutants, e.g., glial subtypes that were affected in *PINK1*[P399L]. The commonly affected cell types include cholinergic olfactory projection neurons (OPNs), Kenyon cells, cholinergic neurons of the visual system (T1 neurons), and other yet-to-be-identified cell types (*Figure 2b–f*, black cells; *Supplementary file 6*). Hence, while our mutants are modeling different

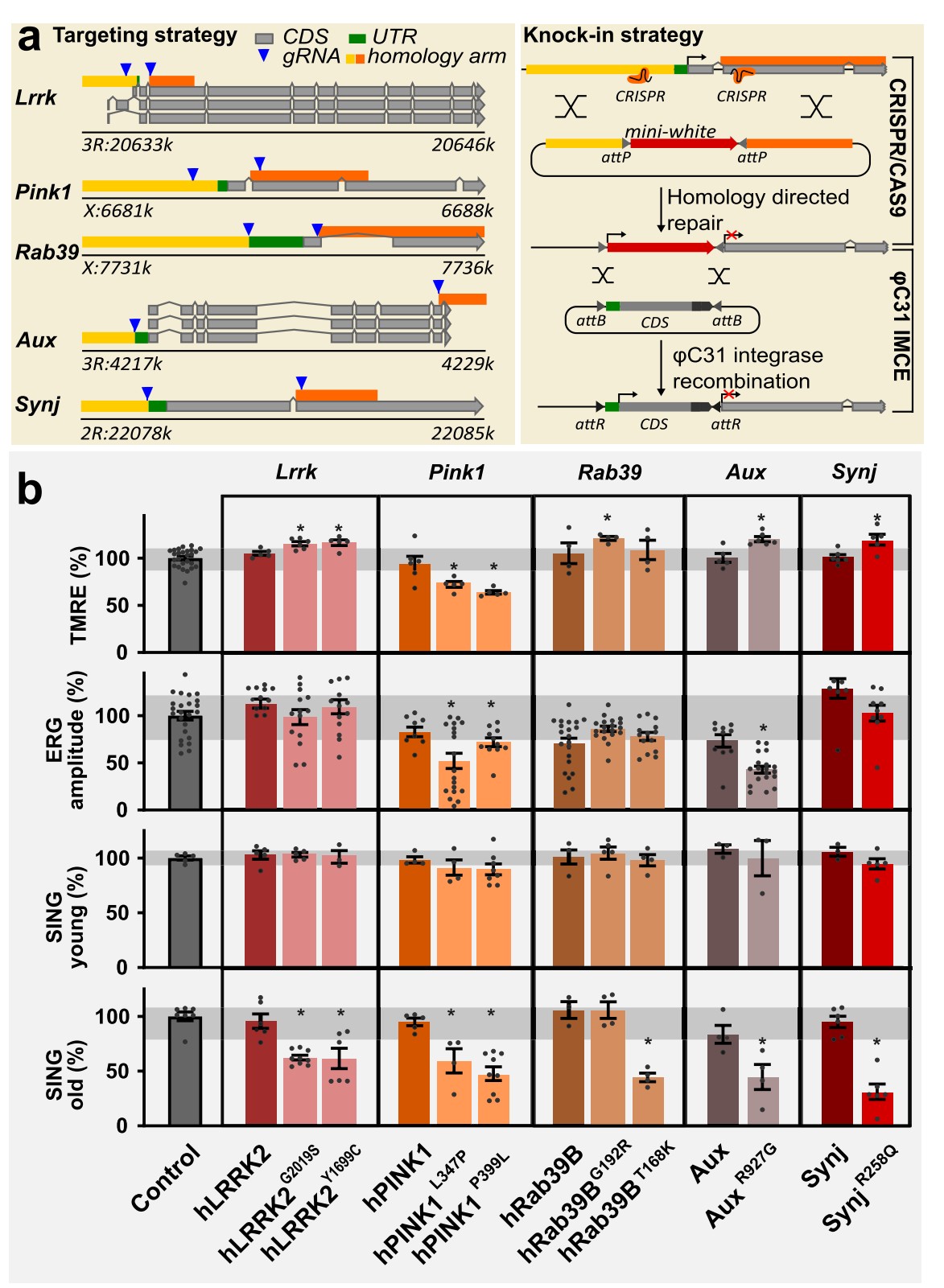

**Figure 1.** A collection of familial Parkinson's disease (PD) knock-in models. (**a**) Scheme of the knock-in strategy where the first common exon in all *Drosophila* transcripts (left) was replaced by an attP-flanked mini-*white* gene using CRISPR/Cas9 mediated homologous recombination, creating a null mutant, and then replaced by wild-type or pathogenic mutant human or fly coding DNA sequence (CDS) using PhiC31 mediated integration (right). The chromosomal positions of the genes are indicated. All knock-ins are in the same isogenic genetic background (cantonized $w^{1118}$). (**b**) Phenotypic analysis

*Figure 1 continued*

of PD mutants: (1) Mitochondrial membrane potential measured by ratiometric tetramethyl rhodamine ethyl ester (TMRE) fluorescence at neuromuscular junction (NMJ) boutons in third-instar larvae. n≥4 animals per genotype and 10 boutons from ≥3 NMJs per animal. (2) Depolarization amplitude quantified from electroretinograms (ERGs), recorded after 7 days of light exposure. n≥20 animals per genotype. (3) Quantification of startle-induced negative geotaxis (SING) at 5±1 days after eclosion (young) and 25±1 days after eclosion (old). 95% of *pink1* mutants died <25 days and were tested at 15 days. Values are normalized to control (see Materials and methods). Variance of control measurements are in gray. Bars are mean ± s.e.m.; *, p<0.05 ANOVA/Dunnett's test.

The online version of this article includes the following figure supplement(s) for figure 1:

**Figure supplement 1.** Characterization of Parkinson's disease (PD) knock-in models.

genetic origins of disease, we find some cell types are commonly affected. While several cell types are affected (and many are cholinergic), OPN are here of particular interest because they are higher-order projection neurons that control innate olfactory processing and odor classification (*Masse et al., 2009*).

## Synaptic genes and pathways are deregulated in cholinergic brain regions of fly PD models and human PD patients

We next compared the identity of the DEG found in PD fly model cholinergic OPNs to the identity of the (homologous) genes differentially expressed in patient brain samples rich in cholinergic neurons. We analyzed *nucleus basalis of Meynert* (NBM), *nucleus accumbens*, and *putamen* from five brains of PD patients (all have idiopathic *LRRK2* risk mutations) and five brains from unaffected age-matched individuals (without PD-relevant genetic variants). Our analysis identified 37 distinct cell types in these brain regions (*Figure 3—figure supplement 1a*), and most of the cell types have a similar frequency when comparing PD versus non-PD condition. An exception is a microglia subtype (MG #20, *Figure 3—figure supplement 1b*), which is 900% increase in frequency in PD patient brains compared to non-PD samples. We next analyzed all neuronal subtypes in these samples (*Figure 3—figure supplement 1c and d*) and identified the DEG. Many of the DEG are genes encoding synaptic proteins that are also listed in the synapse-specific portal SynGo (*Koopmans et al., 2019*; *Figure 3a and a'*; *Supplementary file 7*). We then asked how they compare to the DEG found in affected cholinergic projection neurons of our five fly models (*Figure 3a″*; *Supplementary file 7*). We therefore first listed the deregulated genes in affected cell types for each fly model. There are many DEG unique to each PD mutant, but there is also overlap in DEG between the models. To define this overlap across fly PD mutants, we ranked genes according to their signed differential expression for each of the five PD models (up or downregulated) and then summed their rank across the mutants retaining the genes in the top 5% and the bottom 5% as the common highly up- or downregulated genes (*Supplementary file 7*); i.e., only if the DEGs are mostly commonly up- or commonly downregulated they end up at the top or bottom of our list. Additionally, when we compare this list of deregulated fly genes to the DEG-orthologues in PD patients, there is remarkable overlap. For example, in the NBM, an area associated with PD (*Arendt et al., 1983*), >20% of the neuronal DEG that have an orthologous gene in the fly are also found among the most deregulated genes across PD fly models. This is highly significant: of the 2486 significantly differentially expressed human genes, 1149 have a fly orthologue, and of these, 28.46% overlap with the deregulated fly genes (5% top and bottom genes listed in *Supplementary file 7*). Performing a hypergeometric test confirms that this overlap is significant (p, 9.06e$^{-76}$).

Next, we analyzed the list of human-fly overlapping DEG using Gorilla gene ontology enrichment (*Eden et al., 2009*). Markedly, this shows that the processes most strikingly enriched in both fly models and human NBM are those related to synaptic function (*Figure 3b*). Indeed, more than half of the DEG found in human and flies are present in the synapse-specific portal SynGo (*Koopmans et al., 2019*; *Figure 3a*; *Supplementary file 7*). We found enrichment for presynaptic function, including synaptic exocytosis, axon guidance, synapse organization, ion transport, and protein homeostasis (*Figure 3b and c*). Additionally, we also found enrichment of genes involved in RNA regulation and mitochondrial function that are also important for the functioning of synaptic terminals (*Supplementary file 7*; *Gorenberg and Chandra, 2017*; *Morais et al., 2009*; *Snead and Eliezer, 2019*; *Uytterhoeven et al., 2011*; *Verstreken et al., 2005*). Of note, while the human samples all have *LRRK2* variant mutations, comparing the vulnerable gene signatures from each of the PD fly models to the DEGs from the human samples does not show any greater similarity between the *LRRK2* mutants compared to the

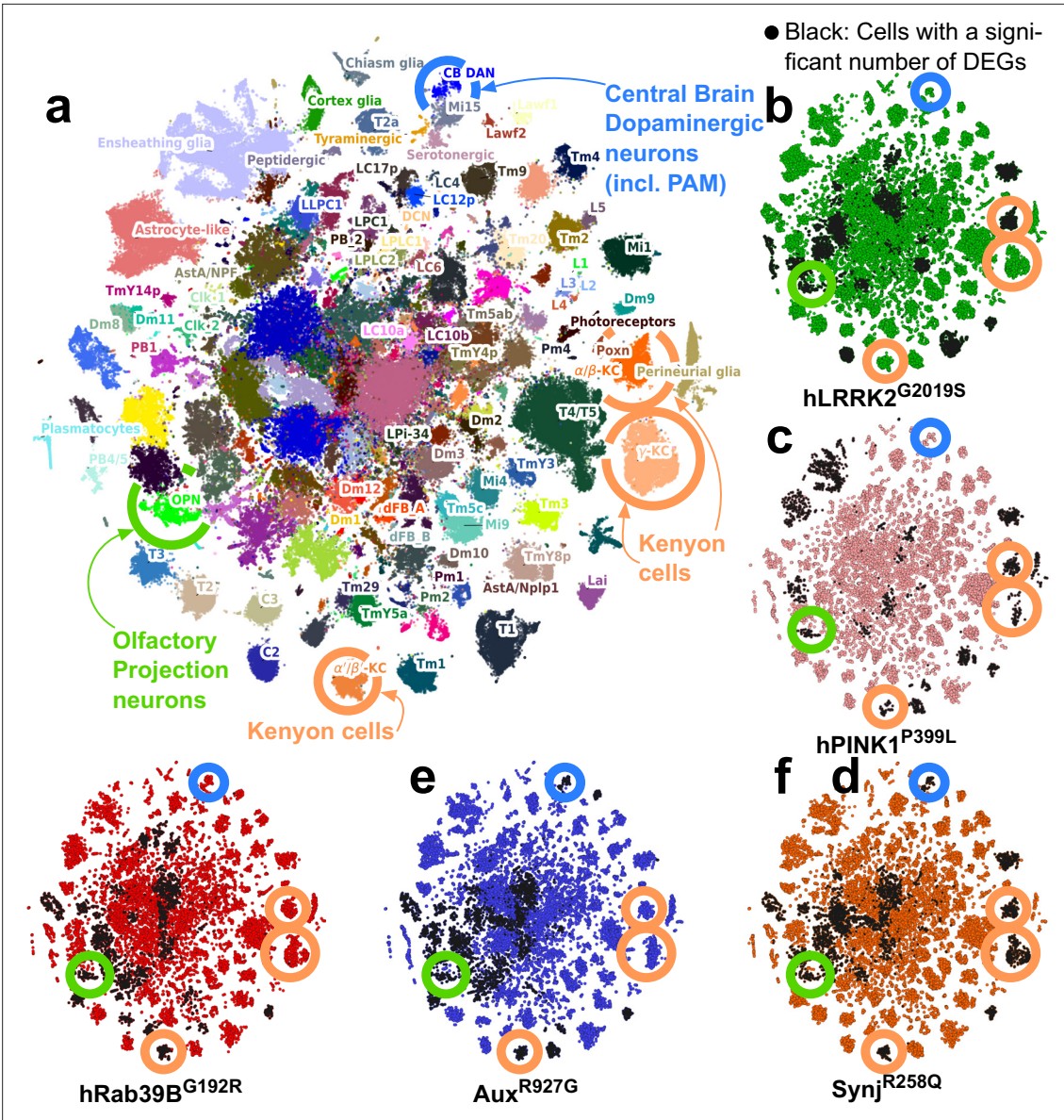

**Figure 2.** Single-cell RNA sequencing reveals that olfactory projection neurons (OPNs) are impaired across young Parkinson's disease (PD) fly models. (**a**) tSNE of 226k cells characterized by single-cell RNA sequencing of entire young (5-day-old) fly brains (controls and PD knock-in mutants). Colors indicate the cell types; 186 of which were identified (see Materials and methods). Key cell types are encircled, including dopaminergic neurons (DAN, blue), mushroom body Kenyon cells (orange), and OPN (green). (**b–f**) tSNE of the cells from five selective PD knock-in mutants (5 days of age). Black cells are those with a significant transcriptomic change. Key cell types labeled in (**a**) are indicated. Note that OPN are consistently affected across mutants, while DAN are not at this early stage.

The online version of this article includes the following figure supplement(s) for figure 2:

**Figure supplement 1.** Validation and analysis of single-cell RNA sequencing data.

other PD mutants (*Supplementary file 7*). In summary, cholinergic olfactory neurons of young familial fly PD models and human PD patients both show preferential transcriptional deregulation of genes that regulate synaptic function.

## Young PD models suffer from synaptic impairment of cholinergic OPN

The transcriptional deregulation of genes encoding synaptic proteins suggests synaptic alterations in the affected cell types. We therefore expressed the $Ca^{2+}$-sensor GCaMP3 in the OPN, one of the majorly affected cell types in our PD flies, and monitored synaptic terminal function through a

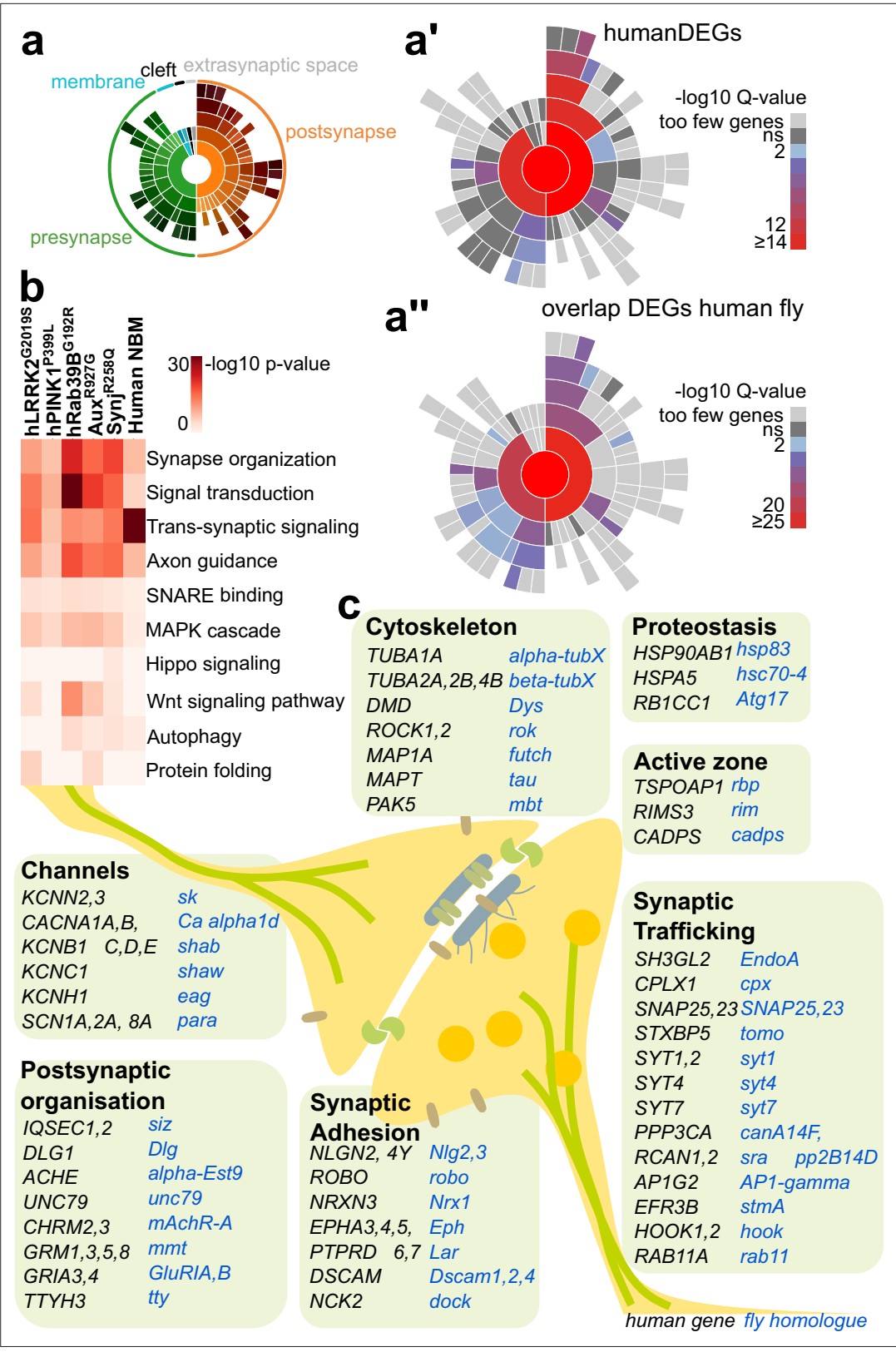

**Figure 3.** Common differentially expressed genes (DEGs) in cholinergic neuron-rich brain regions of human Parkinson's disease (PD) patients and PD fly models. (**a–a″**) Schematic of the sunburst plot indicating Gene Ontology (GO) terms for each sector (**a**) and the mapping of the DEGs in *nucleus basalis of Meynert (NBM)*, *nucleus accumbens,* and *putamen* brain samples idiopathic PD patients (with *LRRK2* risk mutations) and controls

*Figure 3 continued on next page*

*Figure 3 continued*

(**a'**) and mapping of the DEGs found commonly in fly and human samples (**a''**). Inner rings represent the different GO categories (indicated in a), with their subcategories in the outer rings, rings (in a'–a'') are color-coded according to enrichment Q-value. (**b**) GO analysis of DEG in cholinergic neurons of young PD fly models and NBM neurons of PD patients. Redundant terms were removed. Color: adjusted p-value. (**c**) Schematic of the DEGs found commonly in fly PD models (blue) and human PD samples (black) manually sorted according to their previously described synaptic functions. **Supplementary file 7** contains the summarized results of the DEG analysis of fly brains and postmortem human brain samples and the SynGO analysis.

The online version of this article includes the following figure supplement(s) for figure 3:

**Figure supplement 1.** Single-cell RNA sequencing of cholinergic regions in postmortem human brain samples.

small window in the head capsule of living flies upon stimulation (**Figure 4a and b**). Neuronal stimulation results in a robust synaptic Ca$^{2+}$ signal at OPN synapses in controls (**Figure 4c and d**, black points, quantified in **Figure 4g**, gray bar) and in animals with the wild-type PD gene knocked-in (e.g. hLRRK2 in **Figure 4c and d**, dark red points, quantified in **Figure 4g**, left bar). In contrast, the five PD mutant knock-ins show significantly weaker Ca$^{2+}$-responses (e.g. hLRRK2$^{G2019S}$ in **Figure 4c and d**, pink points, and quantified in **Figure 4g**, middle bar; 'OPN>wt CDS -'). Next, we tested if this defect is cell-autonomous and created PD mutant knock-ins expressing the wild-type CDS in their OPN using GH146-Gal4 (**Figure 4g**, right bar; 'OPN>wt CDS +'). Even though this Gal4 driver also drives expression in one pair of inhibitory APL neurons (GABAergic neurons that we did not recover in our single-cell sequencing), it mostly expresses in OPN enabling us to assess PD gene function in OPN (**Li et al., 2017**; **Liu and Davis, 2009**). PD mutants expressing wild-type PD genes using GH146-Gal4 show synaptic Ca$^{2+}$ signals similar to controls, indicating the PD genes are required in OPN of these young animals to maintain robust synaptic function (**Figure 4g**, right bar; 'OPN>wt CDS +'). Note that while *hLRRK2* wild-type knock-in flies are very similar to wild-type controls (**Figure 4g**, right bar), the *hLRRK2$^{G2029S}$* knock-in flies are not rescued by OPN-specific expression of *hLRRK2* (**Figure 4g**, right bar; 'OPN >wt CDS +'). This is in agreement with the *G2019S* mutation being dominant (**West et al., 2005**; **Zimprich et al., 2004**).

Previous work indicated that functional defects of OPN often cause morphological changes at the level of the synaptic area in the calyx (**Kremer et al., 2010**). We thus also assessed the synaptic area of OPN in the calyx (based on the GFP fluorescence area of the GCaMP3 sensor). Interestingly, the PD mutants have a significantly larger area compared to controls, despite very similar GFP expression levels (**Figure 4e, f and h**). Again, this defect is cell-autonomous, because it is rescued by OPN-selective expression of the wild-type CDS, except - again - for *LRRK2$^{G2019S}$* agreeing with it being a dominant mutant (**Figure 4h**). Thus, OPN of young PD mutants display cell-autonomous presynaptic defects: even though the synaptic area is increased, synaptic Ca$^{2+}$ signals are diminished.

## Hyposmia is prevalent in young PD models

While old (25 days) PD mutants show PAM-dependent locomotion dysfunction in the SING assay (**Figure 1b**, SING old), young (5 days) PD mutants do not show SING defects (**Figure 1b**, SING young). These data indicate PAM neurons are functional in young PD mutants. Yet, these young mutants suffer from OPN-synaptic dysfunction (**Figure 4**). We thus wondered if our young PD models have olfactory performance defects. We subjected flies to a choice between two odors (test 1: motor oil and banana *or* test 2: beer and wine). Control and wild-type knock-in flies show a robust odor preference in these two odor tests (**Figure 5a and b**). Conversely, the knock-in flies carrying the pathogenic mutations behave indecisive (**Figure 5a and b**). Hence, young PD mutants show olfactory behavior defects.

We then asked if this defect is cell-autonomous and expressed the relevant wild-type CDS *selectively* in the OPN of the pathogenic knock-in mutant flies. This manipulation restores olfactory performance back to control levels (**Figure 5c and d**; 'OPN>wt CDS +', green label) when compared to PD mutants lacking OPN expression (**Figure 5c and d**; '-'). However, the *LRRK2$^{G2019S}$* mutant remains unaffected, consistent with its classification as a dominant mutant. Note that *lrrk$^{KO}$* flies also show an olfactory performance defect (**Figure 5c and d**; '-') and that expressing *hLRRK2* cDNA specifically in the OPN of *lrrk$^{KO}$* rescues this defect (**Figure 5c and d**; 'OPN>hLRRK2 +', green label). This indicates hLRRK2 is a functional protein that, in this context, can compensate for the function of the fly orthologue. Finally, we demonstrate that the rescue of the olfactory performance in PD mutants is specific

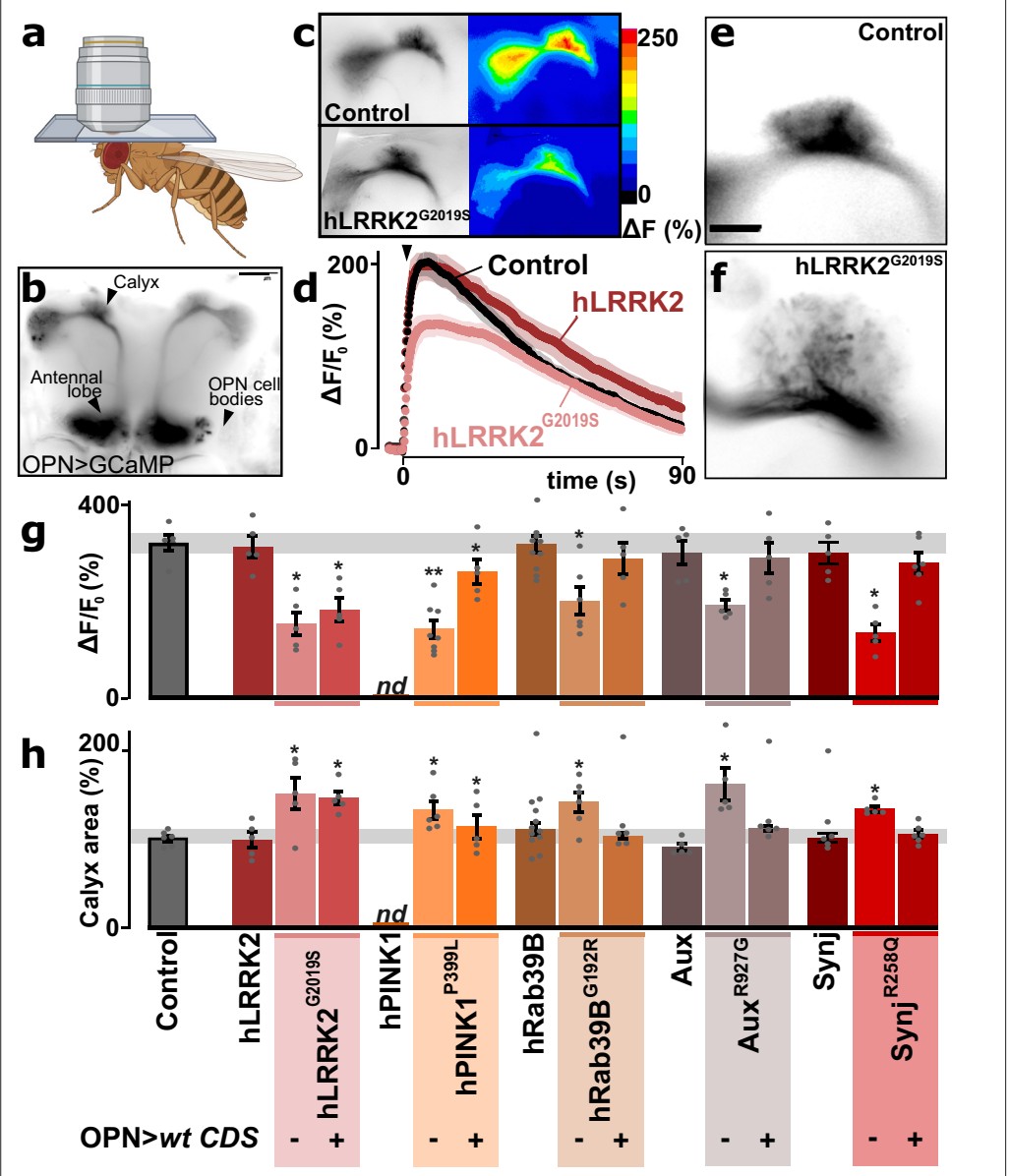

**Figure 4.** Synaptic defects in olfactory projection neuron (OPN) of Parkinson's disease (PD) mutants. (**a**) Schematic of a head-fixed awake fly for live $Ca^{2+}$-imaging through a window in the head capsule. (**b**) Confocal image of GCaMP3-fluorescence expressed in OPN using GH146-Gal4, indicating the locations of the cell bodies, antennal lobe, and the calyx (scale bar: 50 μm). (**c–d**) Confocal image of the stimulus-induced change of fluorescence (peak amplitude) in the synaptic region of the calyx of a control and an *hLRRK2^{G2019S}* knock-in animal (**c**) and quantification of fluorescence change (± SEM) over time (**d**). Arrowhead: time of stimulus application (10 mM nicotine, see Materials and methods). (**e, f**) Images of GFP fluorescence marking the synaptic area of OPN in the calyx of control (**e**) and *hLRRK2^{G2019S}* knock-ins (**f**). Scale bar is 20 μm. (**g**) Quantification of GCaMP3 peak amplitude at OPN synapses in the calyx following stimulation (10 mM nicotine) in controls, wild-type coding DNA sequence (CDS) knock-ins, and in the PD knock-in mutants where the wild-type CDS is not (-) or is (+) expressed in OPN using GH146-Gal4 (OPN>wt CDS). Note that the *hPINK1* control could not be determined as the combination of nSyb-Gal4>UAS-hPINK1 (expression in all neurons) in the *Pink1* knock-out background interferes with the OPN-specific expression of Gal4 to drive UAS-GCaMP3 expression (**g**) (left bar; 'nd'). In contrast, this issue is not present in *hPINK1^{P399L}* mutant knock-in flies (nor the other flies used in the study) that could be rescued by OPN-specific expression of *hPINK1* (**g**) (right bar); ('OPN>wt CDS +'). (**h**) Quantification of the GFP fluorescence area of OPN synapses in the calyx (based on GCaMP3 signal) in controls, wild-type CDS knock-ins, and in the PD knock-in mutants where the wild-type CDS is not (-) or is (+) expressed in OPN (OPN>wt CDS). For (**g, h**) n≥5 animals per genotype. Bars are mean ± s.e.m. *, p<0.05 in ANOVA/Dunnett.

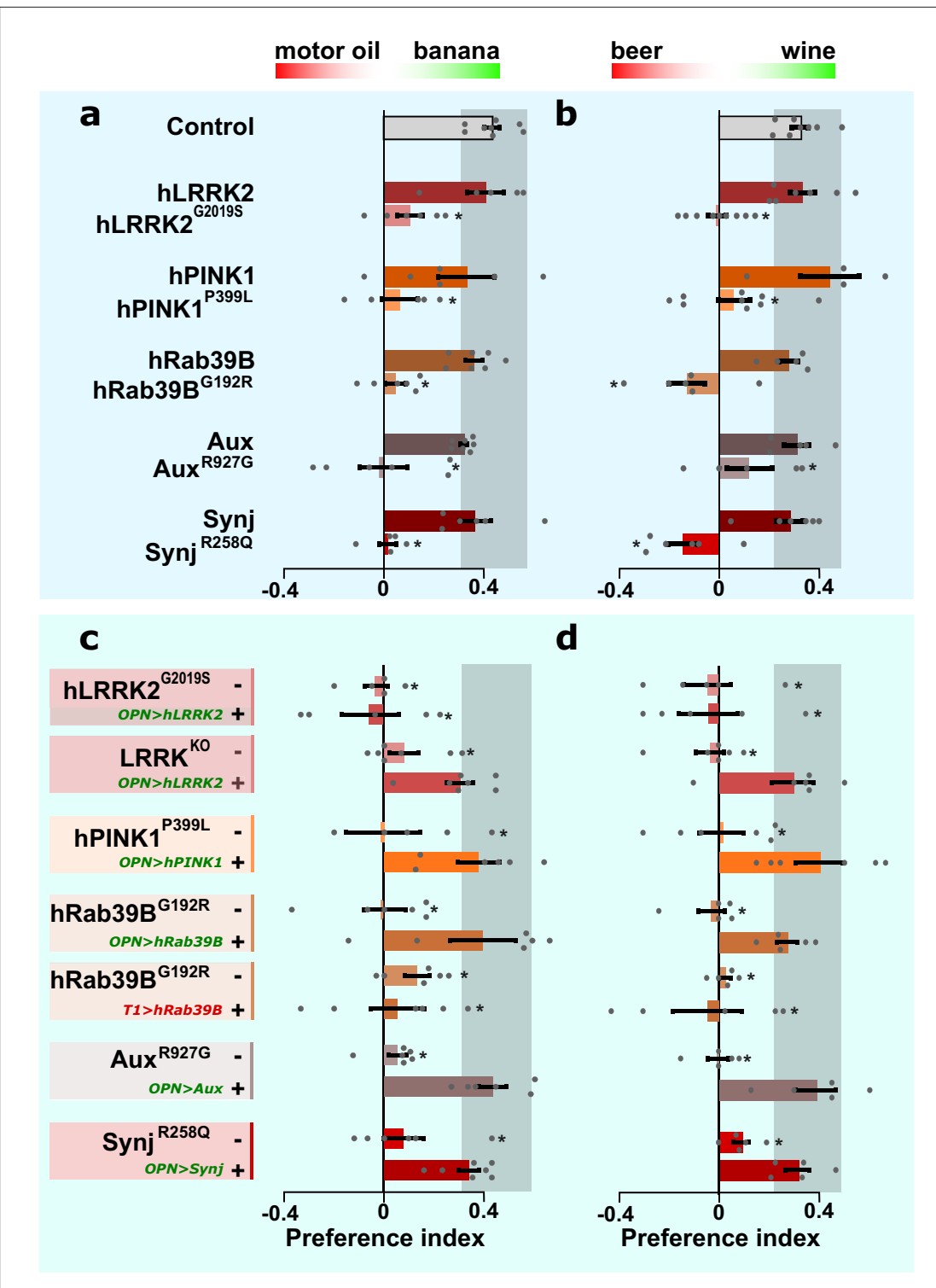

**Figure 5.** Olfactory projection neurons (OPN) dysfunction causes hyposmia in Parkinson's disease (PD) mutants. (**a, b**) Olfactory performance of PD knock-in flies and controls when given the choice between a blend of motor oil and banana odors (**a**) or a blend of beer and wine odors (**b**). (**c, d**) Olfactory performance of PD knock-in flies and *LRRK*$^{KO}$ when the relevant wild-type coding DNA sequence (CDS) is not (-) or is expressed selectively in OPN ('OPN>wt CDS +', green label) or T1 cholinergic interneurons ('T1>hRab39B +', red label), as a negative control. For (**a–d**) bars are mean ± s.e.m. n≥5 assays, ≥200 flies per genotype. *, $p < 0.05$ in ANOVA/Dunnett.

for OPN, because expressing wild-type CDS in another affected cell type, 'T1-interneurons of the visual system', of knock-in PD animals (we chose to test this in *Rab39B*[G192R] mutants) does not improve olfactory performance (*Figure 5c and d*; 'T1>hRab39B +', red label). Hence, a cell-autonomous defect in cholinergic OPN causes olfactory defects across PD mutants.

## Rescuing OPN dysfunction in young PD mutants prevents DAN defects at older age

Our data indicates that young PD mutants (*aux, synj*, and *LRRK2*) have olfactory defects while displaying normal DAN function (SING). Conversely, aged *aux*[R927G], *synj*[R258Q], and *hLRRK2*[G2019S] knock-in animals exhibit both olfactory preference defects and progressive DAN (SING) dysfunction (*Figure 6a*; line graphs; 'DAN-dependent movement behavior'). We also assessed the synaptic afferent innervation area of PAM DAN onto mushroom bodies by quantifying the area of anti-TH labeling. In line with the SING behavioral defect, all three 25-day-old knock-in mutants show a significant reduction in DAN afferent (*Figure 6a*, left bar; 'OPN>wt CDS -'; 'DAN synapse morphology'). We then wondered if rescuing OPN at young age affects DAN at later age. We therefore expressed the wild-type PD gene in (a.o.) OPN of *aux*[R927G] and *synj*[R258Q] mutants using the GH146-Gal4 (that does not drive expression in DAN) (*Figure 6a*, right bar; 'OPN>wt CDS +'; 'DAN synapse morphology'). Given that *hLRRK2*[G2019S] is dominant, we did not express the wild-type *hLRRK2* in OPN, but *endoA*[S75D] that we showed previously genetically interacts with *lrrk* (*Matta et al., 2012*; *Figure 6a*, right bar; 'OPN>endoA[S75D] +'; 'DAN synapse morphology'). We find these genetic manipulations rescue the DAN-associated SING and synaptic connectivity defects in old PD mutants (*Figure 6a*). Altogether, these results suggest that functional OPN have a cell-nonautonomous effect to prevent DAN dysfunction that emerges at old age in the PD mutants.

To further substantiate this observation, we also conducted a set of independent experiments where we fed *hLRRK2*[G20190S] mutants nicotine. Nicotine activates the nicotinic acetylcholine receptors that, under normal circumstances, are also activated by the release of acetylcholine from cholinergic neurons, including OPN (*Chambers et al., 2013*; *Changeux, 2010*; *Zhou et al., 2001*). While feeding nicotine does not rescue the olfactory preference defect of *hLRRK2*[G2019S] mutants, it also does not rescue the OPN synapse morphology defect or the OPN-associated defects in synaptic Ca-imaging (*Figure 6b*). Interestingly, nicotine does rescue the DAN-associated defects, including SING, synapse loss and defects in Ca-imaging at DAN synapses (*Figure 6c*). These results are consistent with the role of OPN function, specifically acetylcholine release, in preventing DAN damage in PD mutants.

In a final experiment we assessed if this effect of nicotine on DAN health is conserved across species. We therefore generated human induced neurons derived from iPSC in which we engineered an *LRRK2*[G2019S] mutation and differentiated the cells into DAN (*Figure 6—figure supplement 1*; *Kriks et al., 2011*; *Nolbrant et al., 2017*). Our protocol results in >50% DAN in both the *LRRK2*[G2019S] mutant line and the isogenic control (*Figure 6d*, *Figure 6—figure supplement 1e and f*). To assess neuronal activity, we expressed GCaMP6f and find that *LRRK2*[G2019S] mutant DAN are significantly less active and the amplitudes of their $Ca^{2+}$-spikes are smaller than those in isogenic controls (*Figure 6e*). Next, we incubated the differentiated DAN with nicotine and found this rescues the *LRRK2*[G2019S]-induced neuronal activity defects. This effect is specific to nicotine as no rescue was observed when cells were co-incubated with mecamylamine, a non-competitive antagonist of nicotinic acetylcholine receptors, trumping the effects of nicotine (*Figure 6e*). Hence, the positive effect of nicotine on PD mutants is conserved between flies and iPSC-derived DAN.

## Discussion

Parkinsonism is currently diagnosed based on motor defects that only occur following significant DAN degeneration. However, patients suffer from non-motoric problems decades earlier, including sleep problems and hyposmia (the inability to properly discern odors) (*Yamakado and Takahashi, 2024*). Most of these non-motoric problems are not responsive to dopamine replacement therapy, suggesting other transmitter systems are involved. Using unbiased methodology, we find cholinergic system failure as one of the earliest defects across five different Parkinsonism fly models and we show that synaptic failure of specific cholinergic projection neurons (OPN) causes non-motor problems, including hyposmia. Also in patients, cholinergic defects in several brain regions have been observed,

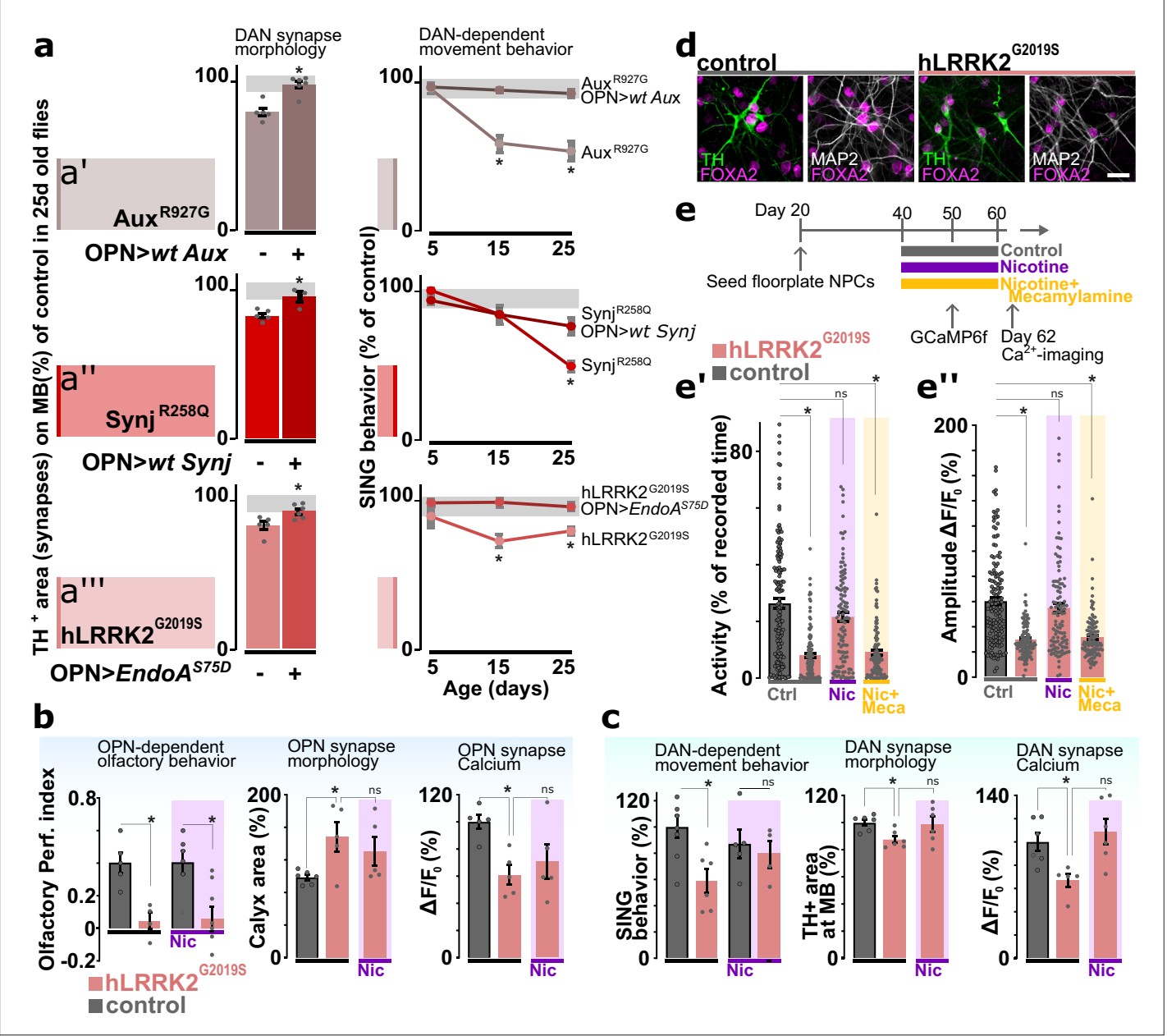

**Figure 6.** Cholinergic neuron activity rescues dopaminergic defects that occur at later stages in the life of Parkinson's disease (PD) mutants. (**a**, left) Synaptic area of dopaminergic neuron (DAN) innervating the mushroom body in aged PD models (Aux^R927G, synj^R258Q, or LRRK2^G2019S) and with or without GH146-Gal4-driven expression of the wild-type PD gene (*aux* or *synj*) or *EndoA^S75D*, respectively. Bars: mean ± SEM. n≥5, *p<0.05 in ANOVA, Dunnett's test. (**a**, right) Startle-induced negative geotaxis (SING) of the PD models with or without GH146-Gal4-driven expression of wild-type gene or *endoA^S75D*. Points: mean ± SEM. n≥5, *p<0.05 in two-way ANOVA. Gray zone: variance of controls. (**b**) Odor choice performance, stimulus-induced changes in synaptic Ca²⁺ signal and olfactory projection neuron (OPN) synapse area of young controls and *hLRRK2^G2019S* flies with or without chronic nicotine (Nic) feeding (up to 1 day before testing). Bars: mean ± SEM. n≥5 assays, *p<0.05 in ANOVA, Dunnett's test. (**c**) SING, stimulus-induced changes in synaptic Ca²⁺ and DAN synapse area of aged controls and *hLRRK^G2019S* flies with or without chronic application of nicotine. Bars: mean ± SEM. n≥5 assays, *p<0.05 in ANOVA, Dunnett's test. (**d**) Confocal images of differentiated (60 days) wild-type and *LRRK2^G2019S* ventral midbrain DAN labeled with the ventral midbrain marker FOXA2, dopaminergic marker TH, and neuronal marker MAP2. Scale bar: 20 µm. (**e**) Scheme of the treatment protocol and spontaneous Ca²⁺ activity (**e′**) and amplitude (**e″**) of human induced DAN, 2 days after 20 days of no treatment (Ctrl), nicotine (Nic) treatment or nicotine+mecamylamine (Nic+Meca) treatment. Bars: mean ± SEM. n≥60 DAN from three independent differentiations, *p<0.05 in ANOVA, Dunnett's test.

The online version of this article includes the following figure supplement(s) for figure 6:

**Figure supplement 1.** Generation and validation of human LRRK2^G2019S ventral midbrain dopaminergic neuron (DAN).

including the olfactory bulb and the occipital cortex (*Kuhl et al., 1996*; *Shimada et al., 2009*). These defects correlate with several non-motor symptoms, including impaired olfactory and visual processing (*Bohnen et al., 2010*). We now place cholinergic failure firmly ahead of dopaminergic system failure in flies, and we suggest that dysfunction of specific cholinergic projection neurons may be a prodromal signal of disease. Our work opens the possibility to explore the early detection of specific cholinergic interneuron dysfunction, e.g., using PET tracing, as an entry point to move diagnosis and intervention ahead of dopaminergic degeneration.

The connection between early cholinergic neuron dysfunction and subsequent failure of the DAN system in disease progression remains uncertain. However, DAN have a high density of nicotinic acetylcholine receptors, and evidence suggests that stimulating these receptors enhances DAN survival (*Brimblecombe et al., 2018*). This is supported by epidemiological data indicating that nicotine exposure from chronic smoking protects against PD (*Janssens et al., 2022*; *Wang et al., 2019*), and various studies, including the work we present here, show that direct nicotine supplementation mitigates DAN dysfunction in mouse and fly PD models (*Chambers et al., 2013*; *Fares et al., 2023*; *Olsen et al., 2023*). We also demonstrate this with a genetics experiment where we express wild-type PD genes in OPN and observe rescue of DAN dysfunction in older flies. These findings imply that factors affecting the function of OPN or cholinergic neurons might, by absence of insufficient innervation or activity, lead to DAN problems and degeneration, warranting further exploration of the underlying molecular mechanisms.

Our findings reveal a common trigger relevant for both familial and idiopathic cases. Observations of Lewy body pathology in postmortem samples suggest the disease may nucleate at specific sites and then propagate. This includes spreading of the disease between the gut and the brain as well as early aggregation of α-Synuclein, a synaptic protein, in the olfactory bulb (*Braak et al., 2003a*). The presence of protein aggregates in these areas is consistent with protein homeostasis and synaptic defects that ultimately contribute to the aggregation of α-Synuclein (*Chu et al., 2009*; *Friedman et al., 2012*; *Sarkar et al., 2007*; *Webb et al., 2003*). In idiopathic disease, there is some evidence this is because the gut and olfactory bulb synapses are more easily exposed to environmental toxins that cause protein misfolding and aggregation (*Braak et al., 2003b*). Furthermore, we also find genetic interactions with the protein turnover machinery that warrant further investigation (this study and *Matta et al., 2012*). However, different mechanisms may be at play since our single-cell sequencing shows the deregulation of several synaptic pathways in the PD knock-in mutants. While several genes mutated in familial forms of PD have proposed synaptic functions (endocytosis, synaptic autophagy, etc.), others may be affecting the synapse indirectly (e.g. mitochondria, transport, etc.). Future work will now need to elucidate which synaptic processes are affected and how this has an impact on cholinergic projection neuron function and cholinergic transmission.

## Strengths and caveats to our approach

### Strengths

In this work, we generated new *Drosophila* PD models using a controlled and standardized design to introduce pathogenic mutant PD variants at the endogenous locus. With this approach, the gene regulatory environment is as much as possible preserved ensuring that our models are physiological relevant, circumventing overexpression artifacts or ectopic effects. By studying these mutations in their endogenous context, we can better understand their specific contributions to transcriptional and cellular phenotypes. Given that the mutants are generated in a controlled 'isogenic' fashion, it is now possible to conduct careful side-by-side comparisons of different PD mutant genotypes.

Analyzing these newly generated PD models at young age using single-cell RNA-seq enables us to identify affected cell populations and uncover early molecular events that might precede overt neurodegenerative disease phenotypes. Furthermore, the comparison between our fly PD models and human patient brain samples provides evidence that some of the molecular mechanisms of disease are conserved across species.

Finally, our in vivo investigations enabled us to uncover cell-non-autonomous effects of cholinergic projection neurons onto the fly dopaminergic system, a feat that would have been difficult to discover in in vitro (mono-) human neuron cultures.

## Caveats

One of the caveats in our study involves the use of the GH146-Gal4 line. GH146-dependent Gal expression includes OPNs (that are cholinergic) and one pair of inhibitory APL neurons (that are GABAergic). There are only two APL per fly brain and our single-cell sequencing experiment does not have the resolution to allow us to test if these neurons had a significant number of DEG. However, we are able to rescue DAN dysfunction by mimicking cholinergic output (using nicotine). These data do not exclude that APL-neuron problems contribute to the defects we observe in our PD mutants, but they do suggest that cholinergic output is critical to maintain normal DAN function.

Another caveat involves the temporal mismatch of comparing single-cell RNA-seq data obtained from young PD models versus postmortem end-stage human disease tissue. Indeed, there may be changes in early stages that may not completely align with the late-stage disease profiles typically observed in human postmortem samples. Unfortunately, obtaining human brain samples from PD patients at an early stage are rare, these samples would have been more relevant to be compared to our young PD models. Nonetheless, we do also find that cholinergic neurons from postmortem brain samples have similar affected DEG pointing to similar affected molecular pathways when comparing to our young PD fly data.

## Materials and methods

### Experimental model details

The 'control' we refer to is a semi-isogenized $w^{1118}$ backcrossed to CantonS for 10 generations (cantonized $w^{1118}$). We sequenced the entire genome of the resulting fly line and defined there are no deleterious mutations in homologues of known causative PD genes or in homologues of human genes located close to GWAS loci. All our knock-ins (see below), UAS (see below), and Gal4 lines were backcrossed for >10 generations to this control. All generated fly lines were verified by multiple PCRs and PCR-product sequencing.

Flies were reared on standard yeast food medium at 25°C and kept on 12/12 hr light/dark cycles. Flies were kept in mixed populations of analogous density and were flipped onto fresh food every 3 days. Brains for RNA experiments, including single-cell sequencing, were dissected between 11 and 12 hr zeitgeber time, behavioral and physiology experiments were performed between 18 and 20 hr zeitgeber time. Males and females were tested together, quantified separately, and results pooled if there was no statistical difference. Male hPINK1 pathogenic mutant knock-in flies are sterile, requiring balancing of the stock and consequently they produce only male hemizygotes to be quantified in experiments.

### Generation of *Drosophila* PD models

A complete list of primers, gRNAs, oligos, gBlocks, and fly strains is listed in *Supplementary file 1*. To clone inserts into plasmids NEBuilder HiFi DNA Assembly (NEB) was used.

Unique gRNAs for *LRRK2*, *PINK1*, *Rab39B*, *Aux*, and *Synj* were identified by https://crispor.gi.ucsc. edu/crispor.py and introduced into pCFD4: U6:1-gRNA U6:3-gRNA (*Port et al., 2014*) by following an established protocol: https://crisprflydesign.org/wp-content/uploads/2023/12/Cloning-with-pCFD4. pdf.

Homology arms of 1 kB surrounding the first common exon of the different transcripts from the gene of interest were cloned into pWhite-STAR (*Choi et al., 2009*). For *Aux*, homology arms surrounded the entire *Aux* gene. HiFi DNA Assembly (NEB) was performed with four fragments: mini-white IMCE-cassette, left homology arm, right homology arm, and the plasmid backbone. The homology arms were either generated by gBlocks (IDT), which were modified in order to make them compatible with the requirements of IDT, or amplified by Q5 PCR from genomic fly DNA of the target genotype (cantonized $w^{1118}$). The mini-white IMCE-cassette and vector backbone were generated by restriction digest on pWhite-STAR (*Choi et al., 2009*) with AvrII and XhoI. In order to avoid cutting of the donor plasmid, the PAM sequence of the target was mutated, if located in the homology arms. Mostly, NGG to NAA modifications were used, however, different modifications were used when necessary (presence of PAM in coding region). The donor plasmids pDoC-mini-white-gene (Donor cassette) contains an Integrase mediated exchange (IMCE) cassette surrounded by two homology arms that allows for HDR (homology directed repair). HDR occurs when two double strand breaks are introduced in the

DNA surrounding the first (or second) common exon of all transcripts. This HDR replaced the first common exon by the IMCE-cassette. Site of HDR was directed by the choice of the homology arms in a way such that the two attP sites for IMCE were situated inside evolutionary non-conserved regions.

Rescue plasmids were used to exchange the mini-white IMCE-cassette with the CDS of target genes. All rescue plasmids contain: (1) a stretch of genomic sequence that will be referred to as 'utr': this sequence was deleted together with the removed exon of the target gene and now is being placed back to repair the locus; (2) CDS of the target gene; and (3) a poly(A)-tail. These three elements are surrounded by two attB sites. The rescue plasmids are named pReC-*gene* or pReC2-*gene* and are target gene-specific. The backbone: pReC was generated by linearizing pUC19 with SapI and EcoRI followed by insertion of two gBlocks: attB-MCS-L and MCS-attB-R. We also created pReC2, an optimized version that allows for easier cloning of the utr and CDS as it already contains the SV40 poly(A), by linearizing pUC19 with SapI and EcoRI followed by insertion of two gBlocks: attB-2xMCS-L and 2xMCS-attB-R.

Note that while for *LRRK2, RAB39B,* and *PINK1* knock-in models we used human CDS, for *aux* and *synj* we resorted to *Drosophila* CDS, since knock-in of the human homologues at the locus of these genes did not rescue its loss-of-function (lethality).

## LRRK2

pReC was linearized with SapI. gBlock LRRK2:utr-P2A-SapI-SV40, containing the utr a P2A site (since we knocked-out exon 2), SapI restriction site (for subsequent cloning of the CDS), and an SV40 poly(A), was introduced. The resulting plasmid was linearized with SapI. Next, the CDS of *LRRK2* was amplified from 2XMyc-LRRK2-WT with F_hLRRK2_cDNA and R_hLRRK2_cDNA and introduced into the linearized plasmid. The resulting plasmid is called pReC-hLRRK2. pReC-hLRRK2G2019S and pReC-hLRRK2Y1699C were created by using Q5 Site-Directed Mutagenesis (NEB) with primers F_mut_LRRK2_G2019S, R_mut_LRRK2_G2019S and Fmut_LRRK2_Y1699C, R_mut_LRRK2_Y1699C respectively on pReC-hLRRK2.

## PINK1

pReC was linearized with SapI. gBlock: PINK1:utr-SapI containing the utr and SapI restriction site (for subsequent cloning of the CDS) is cloned into pReC. The resulting plasmid was linearized with AflIII. Next, the gBlock PINK1:SV40poly(A) containing the human Sv40 poly(A) was introduced into the linearized plasmid. This plasmid is linearized with SapI. Next, the CDS of human *PINK1* was amplified from pLenti6-DEST PINK1-V5 WT with F_hPink1_cDNA and R_hPINK1_cDNA and introduced into the linearized plasmid. The resulting plasmid is called pReC-hPINK1. pReC-hPINK1R145Q and pReC-hPINK1L347P were created by using Q5 Site-Directed Mutagenesis (NEB) with primers F_mut_PINK1_L347P, R_mut_PINK1_L347P and F_mut_PINK1_P399L, R_mut_PINK1_P399L respectively on pReC-hPINK1. While we created wild-type CDS knock-ins for rescue experiments and knock-ins carrying the pathogenic mutant variants of all PD models (including the *Pink1* mutant knock-ins), we were unsuccessful in targeting the wild-type human *PINK1* CDS to the fly *pink1* locus, unlike the two pathogenic variants. Therefore, the *Pink1* control we used was generated by neuronally expressing human *PINK1* with UAS-hPINK1 and nsyb-Gal4 (*Yang et al., 2006*) in the *Pink1^{KO}* background (full genotype: *w^{1118} Pink1^{K.O};UAS-hPINK1/+; nSyb-Gal4/+*).

## Synj

pReC was linearized with SapI. gBlock Synj_utr-SapI containing the utr and SapI restriction site (for subsequent cloning of the CDS) is cloned into pReC. The resulting plasmid was linearized with AflIII. gBlock Synj:SV40poly(A) containing the human Sv40 poly(A) was introduced into the linearized plasmid. This plasmid is linearized with SapI. Next, the CDS of *Drosophila synj* and *synj^{R258Q}* were amplified from respectively UAS-Synj+ and UAS-SynjRQ (*Vanhauwaert et al., 2017*) with FW Synj and RC Synj and are introduced into the linearized plasmid by utilizing HiFi assembly. The resulting plasmids are called pReC-dSynj and pReC-dSynjR258Q.

### Rab39B

pReC2 was linearized with XhoI and XbaI. A gblock: Rab39_utr was inserted into the pReC2. Next, this plasmid was linearized with SapI and a gBlock containing the human *RAB39B* CDS was cloned. The resulting plasmid is called pReC2-hRab39. pReC2-hRab39T168K and pReC2-hRab39G192R were created by using Q5 Site-Directed Mutagenesis (NEB) with primers F_mut_hRab39_T168K, R_mut_hRab39_T168K and F_mut_hRab39_G192R, R_mut_hRab39_G192R respectively on pReC2-hRab39.

### Aux

The entire *Drosophila auxilin* gene region was cloned from BAC CH322-22D05 into pRec using BbsI, with an HA-tag at the N-terminal of the gene. AuxR927G was produced by Q5 Site-Directed Mutagenesis (NEB). See also *Jacquemyn et al., 2023*.

### UAS-aux

pUAST.attB was linearized with XhoI and XbaI; gBlock: Aux was cloned into this plasmid. See also *Jacquemyn et al., 2023*.

Constructs were injected in-house or by BestGene (BestGene Inc, CA, US).

## Quantitative RT-PCR

For each sample, 20 heads of 5-day-old males were collected, RNA was extracted using Maxwell RSC instrument and Maxwell RSC RNA kit (Promega) and transcribed using the SuperScriptIII synthesis system (Thermo Fisher Scientific). Subsequently, cDNAs were quantified by qPCR using a LightCycler 480, 480 SYBR Green master mix (Roche) using qPCR primers enlisted in *Supplementary file 1*. mRNA levels were determined using the Δ-Δ-CT method, where Ct values were first normalized to the housekeeping gene Rp49, and next expressed as a percent of endogenous *Drosophila* gene expression.

## Mitochondrial membrane potential (TMRE) assay

Measurements of mitochondrial membrane potential were performed with the potentiometric dye tetramethyl rhodamine ethyl ester (TMRE). Male third-instar larvae were dissected in HL3 (in mM: 110 NaCl, 5 KCl, 10 $MgCl_2 \bullet 6H_2O$, 10 $NaHCO_3$, 30 sucrose, 5 trehalose, and 10 HEPES, pH 7.2). Larval fillets were incubated for 15 min in the presence of 50 nM TMRE (Abcam). Subsequently, the TMRE solution was discarded and fillets were rinsed three times with normal HL3 without TMRE. Mitochondrial labeling of TMRE was imaged on a Nikon spinning disk confocal microscope with a ×40 water dipping objective 0.8 NA, excitation wavelength of 561 nm, and an emission filter 595/50 nm. We imaged at 0.5 µm Z-steps entire NMJ branches. Larvae were stained a mutant and control larva were processed and imaged side-by-side. TMRE labeling intensity was quantified in individual synaptic boutons using Fiji (*Schindelin et al., 2012*). Because of variability between sessions, every mutant was normalized to the controls stained and measured in the same session.

## Electroretinograms

ERGs were recorded as previously described (*Slabbaert et al., 2016*). Flies were immobilized on glass microscope slides, by use of liquid Pritt glue. For recordings, glass electrodes (borosilicate, 1.5 mm outer diameter) filled with 3 M NaCl were placed in the thorax as a reference and on the fly eye for recordings. Responses to repetitive 1 s light stimuli were recorded using Axosope 10.7 and analyzed using Clampfit 10.7 software (Molecular Devices) and Igor Pro 6.37. To assess the ERG response after exposure to constant light, 3-day-old flies were placed under continuous illumination (1300 lux) for 7 days at 25°C prior to ERG data acquisition as described (*Soukup et al., 2016*; *Vanhauwaert et al., 2017*).

## Startle-induced negative geotaxis

Geotaxis locomotion was quantified in mixed populations of ~20 flies (*Benzer, 1967*) with modifications described by *Inagaki et al., 2010*. Flies were flipped into the apparatus without anesthesia and allowed to adjust for some minutes to experimental environment (24°C, 50% humidity). They were tapped down into the first lower tube and allowed to climb into the upper tube for 30 s, at which time

the flies that reached the upper tube were moved to tube 2. The procedure was repeated five times and the SING scored: (#flies in tube 1 + (#flies in tube 2 * 2) + (#flies in tube 3 * 3) + (#flies in tube 4 * 4) + (#flies in tube 5 * 5))/(#total flies * 5).

## Modeling to define the number of cells to be sequenced

We randomly sampled between 0 and 50,000 cells, in steps of 100, from the annotated brain cell dataset generated in *Davie et al., 2018*. For each sample we calculated the number of cell types that were retrieved for different thresholds of detection. This process was repeated 50 times for convergence and Michaelis-Menten kinetics were fit through the data points.

## Brain dissection for single-cell sequencing

30 fly brains (15 females and 15 males, except *Pink1* mutants, where only males were used) were dissected on ice and collected in 100 μl ice-cold DPBS solution, centrifuged at 800 × *g* for 5 min, and placed into 50 μl of dispase (3 mg/ml, Sigma-Aldrich, D4818-2mg) and 75 μl collagenase I (100 mg/ml, Invitrogen, 17100-017). Brains were dissociated in a Thermoshaker (Grant Bio PCMT) for 2 hr at 25°C and 500 rpm. The enzymatic reaction was reinforced by pipetting every 15 min. Subsequently, cells were washed with 1000 μl ice-cold DPBS solution and resuspended in 400 μl DPBS 0.04% BSA. Cell suspensions were passed through a 10 μM pluriStrainer (ImTec Diagnostics, 435001050), cell viability and concentration were assessed by the LUNA-FL Dual Fluorescence Cell Counter, and cells were immediately used for the 10× run. Samples from the different genotypes were separately prepared and sequenced, but they were all processed in parallel to avoid batch effects. The control line (cantonized $w^{1118}$) and two extra wild-type strains (DGRP-551 and $w^{1118}$) were included.

## 10x Genomics high-throughput sequencing

Before sequencing, the fragment size of every library was analyzed on a Bioanalyzer high sensitivity chip. The libraries were diluted to 2 nM and quantified by qPCR using primers against p5-p7 sequence. 10x libraries were sequenced twice: shallow on NextSeq500 (Illumina) to determine library quality and more in depth using NovaSeq6000 (Illumina). The targeted saturation for each sample was around 60%, and additional sequencing runs were performed if needed. Each sequencing run used the following sequencing parameters: 28 bp read 1–8 bp index 1 (i7) – 91 bp read 2.

## 10x Data Preprocessing

The 10x fly brain samples were each processed (alignment, barcode assignment, and UMI counting) with Cell Ranger (version 3.0.2) count pipeline. The Cell Ranger reference index was built upon the third 2017 FlyBase release (*Drosophila melanogaster* r6.16) (*Gramates et al., 2017*). To count the human mutated genes, sequences from the construct were added as artificial chromosomes to the index. For every sample all sequencing runs were aggregated using Cell Ranger and the *–recovered-cells* parameter was specified as 5000.

## Data filtering and clustering

Cell Ranger generated a gene expression matrix, which was used as input for VSN pipelines (GitHub, v 0.27.0) (https://github.com/vib-singlecell-nf/vsn-pipelines; *Flerin et al., 2021*) for the first filtering and data cleanup. These pipelines are written in Nextflow DSL2 making them highly reproducible, and the config file used for the fly samples can be found in *Supplementary file 2*. The VSN pipeline removed predicted doublets using scrublet (*Wolock et al., 2019*), followed by a cleaning of ambient RNA reads using DecontX (*Yang et al., 2020*), similar to the procedures used in the Fly Cell Atlas (*Li et al., 2022*).

Afterward the data was loaded into Scanpy (1.8.2, *Wolf et al., 2018*). Cells were filtered using min_genes = 200, max_genes = 7000, min_counts = 500 and max_counts = 30,000, and finally with percentage of mitochondrial counts below 15. The data were log-normalized with a scale factor of $10^4$ and scaled (maximum values = 10), regressing out the number of UMIs and the percentage of mitochondrial genes as latent variables using a linear regression model. Highly variable genes were selected using sc.pp.highly_variable_genes with default parameters.

We used sc.tl.pca to calculate 250 principal components on the highly variable genes, from which the first 100 components were selected for follow-up analysis based on an elbow plot. Since the data contains many samples (n=65) covering a variety of different ages and genotypes, we noticed

batch effects and low mixing with default parameters. Therefore, we used Harmony (*Korsunsky et al., 2019*) to correct the principal components. For this we used sce.pp.harmony_integrate with max_iter_harmony = 20, using the genotype + age combination as batch variable. The batch effect-corrected components were then used as input to calculate neighbors using sc.pp.neighbors with n_neighbors = 30, followed by Leiden clustering for resolutions between 2 and 20. Subsequently, tSNE and UMAP visualizations were calculated.

### Label transfer

In the previous step different resolutions were used in clustering, leading to an increasing number of clusters detected. To start annotating these and to find a final clustering resolution, we compared the detected clusters with the annotations from the co-clustered published dataset (*Davie et al., 2018*) and annotations from the Fly Cell Atlas (*Li et al., 2022*). In addition, we also used the neural net classifier (*Davie et al., 2018*) using default parameters as described. Finally, we compared the clusters with sorted cell types from *Davis et al., 2020*, using lasso regression as described below. These data were used to annotate clusters in resolution 20, after which clusters with the same annotation were merged. For unknown cells, we chose to use Leiden resolution 5 as basis annotation to not split clusters too much and keep enough cells to perform meaningful differential expression analysis.

In total we found 188 clusters, of which 83 were annotated (median size = 1011 cells, total = 140,778 cells) and 105 unknown (median size = 810 cells, total = 145,010 cells).

### Lasso regression

To expand the published atlas, we analyzed the bulk transcriptomes of multiple FAC-sorted neuronal cell types (*Davie et al., 2018*). Regression models have been previously used to compare bulk transcriptome profiles to aggregated cluster profiles, including non-negative least squares (*Davie et al., 2018*). Here, we use lasso regression to predict single-cell cluster transcriptome profiles using the bulk profiles as variables. Therefore, each single-cell cluster is decomposed as a weighted sum of bulk profiles. The regularization of the lasso model allows for a higher signal-to-noise ratio and clearer assignments. Clear matches between a cluster and a bulk profile are detected as high coefficients (>0.1) in the sum. The regression model was fit using gene signatures derived from Scanpy as features. Before fitting, all datasets were scaled to counts per million CPM using the common genes, independent of previously applied normalization techniques. The calculations were performed using the SciPy (*Virtanen et al., 2020*) and Scikit-learn packages in Python.

### Human brain samples

Usage of postmortem human brain samples was ethically approved (EC reference NH019 2019-02-01). All tissue samples were obtained from the Parkinson's UK Brain Bank at the Imperial College London. The PD cases used in this study all carried a mutation in the *LRRK2* gene (P1542S, M1646T, or R1325Q). One case carried a mutation in *PARKIN* in addition and one case carried a mutation in *EIF4G* in addition. Control samples are from cases without pre- or postmortem PD-associated pathology and diagnosis.

### Human nuclei isolation

Nuclei of human samples were prepared as described in *Hodge et al., 2019*; *Slyper et al., 2020*.

### Calling SNPs

In the human data, 10 different genotypes (5 PD cases, 5 controls) were mixed across 3 regions. To perform demultiplexing, the SNPs are required per sample. Since no genotype information was available, we performed bulk RNA-seq on all samples and called SNPs on the transcriptome. For this we used BCFFtools (v1.11, *Li et al., 2022*). We performed a pileup using the common loci of the 1000 Genomes project (*Auton et al., 2015*; *Gramates et al., 2017*) on the Hg38 iGenomes genome (-d 8000, -Ou), calling SNPs with the bcftools call function (-mv, -Ob). Using the reheader function we then updated the bcf file to make it compatible with demuxlet. We detected 146,848 SNPs, ranging between 25k and 51k per genotype (see table below). Next, the bcf file was sorted in the same order as the bam files and bcftools + fill tags was used to add frequencies.

| Genotype | SNPs |
|----------|------|
| C23 | 25,222 |
| C45 | 41,379 |
| C72 | 37,607 |
| C73 | 40,651 |
| C87 | 48,896 |
| PD106 | 38,281 |
| PD188 | 62,987 |
| PD194 | 46,071 |
| PD348 | 44,195 |
| PD423 | 51,902 |

## Demuxlet

Bam files were filtered using SAMtools (1.11, *Li et al., 2022*) to only contain reads overlapping with the vcf file. Then, popscle dsc-pileup was used to create a pileup file for input in popscle demuxlet (*Kang et al., 2018*). All cells labeled as doublets were removed (~4–7%), and cells assigned to singlets were kept (25–43%). Additionally, a large proportion of cells (~49–69%) were labeled as ambiguous. These cells were assigned to their most likely singlet identity and kept in the remaining analysis. In a next step, we only kept possible genotype-experiment combinations, ending up with 97,489 cells.

## Data filtering and clustering

Cell Ranger generated a gene expression matrix of 151,368 cells. After removing doublets and wrongly assigned cells, 97,489 cells remained. The data was loaded into Scanpy (*Wolf et al., 2018*) and further filtered using min_genes = 500, max_genes = 7000, min_counts = 1000 and max_counts = 30,000, and finally with percentage of mitochondrial counts below 5. We lowered the allowed percentage of mitochondrial counts since the human data was obtained from snRNA-seq. The data were log-normalized with a scale factor of $10^4$ and scaled (maximum values = 10), regressing out the number of UMIs and the percentage of mitochondrial genes as latent variables using a linear regression model. Highly variable genes were selected using sc.pp.highly_variable_genes with default parameters.

We used sc.tl.pca to calculate 200 principal components on the highly variable genes, from which the first 75 components were selected for follow-up analysis based on an elbow plot. We again used Harmony (*Korsunsky et al., 2019*) to correct the principal components for batch effects, using genotype as the batch variable. The batch effect-corrected components were then used as input to calculate neighbors using sc.pp.neighbors with n_neighbors = 30, followed by Leiden clustering for resolutions between 1 and 10. Subsequently, tSNE and UMAP visualizations were calculated.

Using marker genes for main neuronal and glial populations, we chose to combine Leiden resolution 1 for glia and Leiden resolution 3 for neurons.

Gene set enrichment was performed using the AUCell function from the pySCENIC package (0.10.4) in Python. The genes detected in affected cell types of *Drosophila* models were converted to human homologues using the FlyBase database, selecting homologues with a DIOPT score ≥ 5. Significant enrichment was determined using a two-tailed t-test.

## Differential expression

For every cluster, cells from each mutant genotype were compared against cells from the wild-type control. In the fly, OPN subtypes were merged before calculating DEGs. To control for biases in methods, we used Wilcoxon and DESEQ2 (*Love et al., 2014*). The Wilcoxon test was performed with the Scanpy wrapper sc.tl.rank_genes_groups. For DESEQ2 we used rpy2 to call the R package DESEQ2 in Python. DESEQ2 was originally designed to be used on bulk RNA-seq data. Therefore, we grouped cells into pseudobulks based on their experimental run by summing their gene expression profiles. Only pseudobulks consisting of at least 10 cells were kept. We did not select any additional variables to regress.

To detect any method biases, we calculated the Spearman correlation of the signed p-value (log-foldchange multiplied with log(p-val)), showing that both methods differ in the exact number of DEG, but agree quite well in their ranking of up- and downregulated genes. We found larger differences in the number of DEG called in the human data (0–10 in DESEQ2, 200–1800 in Wilcoxon) but the rankings remained highly correlated (r~0.8). The Wilcoxon test is more sensitive to genes expressed in all cells in the cluster, while DESEQ2 can be oversensitive to genes expressed in only a minor fraction of cells.

Additional runs were made in which the number of cells per cluster was downsampled to correct against cluster size-based biases.

## Modeling of DEG

Since the number of DEG has been shown to be highly correlated with the number of cells in single-cell RNA-seq, we used a modeling approach to find outlier clusters with more than expected DEG. Since the data is over dispersed count data, we fit a negative binomial model to the data, with number of differential genes (nDEG) as dependent variable and number of cells in the mutant and number of cells in the control as independent variables:

$$E\left[y_i \mid \alpha, \mu_i\right] = \mu_i$$
$$VAR\left[y_i \mid \alpha, \mu_i\right] = \mu_i + \alpha\mu_i^2$$
$$\log\left(\mu_i\right) = Intercept + B_1 * \log\left(\min\left(X_{1i}, X_{2i}\right)\right)$$

with $y_i$ being the number of differential genes detected in cluster i, $X_{1i}$ the log of the number of cells of cluster i in the mutant, and $X_{2i}$ the number of cells of cluster i in the wild-type.

Models were fit using the statsmodels package in Python and goodness of fit was measured using the log-likelihood (**Supplementary file 4**). The number of UMIs per cluster was not included as independent variable, since it has been implicated with cell size and thus could remove biological effects. Residuals were calculated as the difference between the measured number of differential genes and the predicted number, wherein a positive residual means that the cluster has more differential genes than expected.

$$r_i = \widehat{\mu}_l - y_i$$

## Olfactory behavior assay

Olfactory behavior was quantified in mixed populations of ~50 5±1-day-old flies. Flies were flipped into the experimental setup without anesthesia and allowed to adjust for some minutes to the experimental environment (24°C, 50% humidity). They were placed into a custom-built T-Maze (modified from **Tully and Quinn, 1985**), where they were allowed to freely walk between two tubes for 1 min. Throughout the test, an aquarium pump delivers two separated, constant airstreams (1 l/min) which are directed through an odor vial before entering the tube. Odor vials contained: 2 ml motor oil, 2 g fresh banana, 2 ml undiluted beer (Jupiler), or 2 ml wine (LesTruffiers, 2015) diluted 1:1000. Afterward the olfactory preference index was scored: (#flies in tube with odor A - #flies in tube with odor B)/#total flies. Tests were pseudo-randomized by switching the position of odor tubes every trial. Control flies were tested in each experimental session.

## In vivo Ca²⁺-imaging and synaptic area

5±1-day-old (unless noted otherwise) male flies were placed into a custom-build chamber and the brains were imaged at 5 Hz through a window in the head capsule using a wide field fluorescence microscope (Nikon) equipped with a Hamamatsu camera and a ×20/NA = 0.9 water-immersion objective. 100 mM nicotine was injected into the Ringer's solution covering the brain (final concentration ~10 mM). Images were analyzed using Fiji (**Schindelin et al., 2012**). Fluorescence intensity was quantified in a region of interest (ROI, d=10 μm) that was placed in the center of the synaptic area (calyx for OPNs, mushroom body lobes for dopaminergic PAM neurons) and normalized by subtracting background fluorescence. Basal fluorescence $F_0$ in the synaptic region was determined by averaging five frames before stimulus onset and the change of fluorescence over time was calculated by $F-F_0/F_0$. Flies of related genotypes and controls were always measured within the same sessions, in alternating order.

The area of the calyx was determined by measuring the (2D) surface of the entire visible fluorescent area.

## Nicotine treatment in flies

Nicotine (N3876, Sigma) was mixed with fly food to reach a final concentration of 0.2 µl/ml. A fly weighs ~0.1 mg (*Wu et al., 2016*) and consumes ~1 µl food a day (*Deshpande et al., 2014*). This corresponds to 0.002 µg nicotine uptake per mg body weight daily and corresponds to the amount a smoker absorbs with 10 cigarettes a day. Fly vials containing nicotine and their controls were kept in constant darkness, as nicotine is light-sensitive.

## Generation of LRRK2$^{G2019S}$ hiPSCs

hiPSCs from the KOLF2-1J cell line, obtained from the Jackson Laboratory under the iPSC Neurode-generative Disease Initiative (*Pantazis et al., 2022*), were used as control iPSCs (control, LRRK2$^{WT}$). LRRK2 p.G2019S (LRRK2$^{G2019S}$) was engineered in the KOLF2-1J iPSC line, by means of CRISPR/Cas9 gene editing following *Skarnes et al., 2019*. Briefly, KOLF2-1J iPSCs were nucleofected using the Lonza 4D Nucleofector with Cas9 HiFi protein (IDT), a synthetic guide RNA (Synthego) and a single strand oligodeoxynucleotide (ssODN; IDT) as a donor template (see *Supplementary file 1*). Four days post-transfection, cells were dissociated to single cells and seeded at low density in 10 cm plates. Single colonies were isolated in 96-well plate and sampled for molecular analysis by PCR followed by Sanger sequencing (see *Supplementary file 1*). More than 90% of the colonies harbored the LRRK2 p.G2019S mutation in homozygosity. Selected clones were karyotyped by means of comparative genomic hybridization (CGH) and stained for pluripotency markers. Cell lines were routinely tested for mycoplasma contamination and confirmed to be mycoplasma free before experimental use. The hiPSC lines used in this study are listed in *Supplementary file 1*.

To build the transfer vector pLenti-hSynI-mRuby2-T2A-GCaMP6f (Addgene plasmid #197595), the hSynI-mRuby2-T2A-GCaMP6f-Wpre fragment was PCR-amplified (see *Supplementary file 1*) from pAAV-hSyn1-mRuby2-GSG-P2A-GCaMP6f-WPRE-pA, which was a gift from Tobias Bonhoeffer & Mark Huebener & Tobias Rose (Addgene plasmid #50943; *Rose et al., 2016*) and was introduced into a regular third-generation lentiviral backbone.

## hiPSC differentiation to ventral midbrain DAN

hiPSCs are first differentiated to ventral midbrain neural progenitor cells before they are terminally matured to ventral midbrain DAN using established protocols (*Kriks et al., 2011*; *Nolbrant et al., 2017*). In summary, on day –1, 400,000 hiPSCs/cm$^2$ were seeded on Matrigel-coated wells of six-well plates in StemFlex medium supplemented with 10 µM Rho kinase inhibitor (RI). On day 0, medium was switched to a 1:1 mix of DMEM/F-12 and Neurobasal supplemented with 0.5× N2, 0.5× B27 without vitamin A, GlutaMAX, Penstrep, nonessential amino acids, β-mercaptoethanol (10 µM) (all Life Technologies), insulin (5 µg/ml, Sigma), LDN193189 (500 nM, Sigma), SB431542 (10 µM, Tocris), SHH-C24II (200 ng/ml, Miltenyi Biotec), and Purmorphamine (1 µM, Sigma). CHIR99021 (1.5 µM, STEMCELL Technologies) was added to the medium from day 3 to day 13. SB431542, SHH-C24II and Purmorphamine were withdrawn from the medium at day 9. FGF8b (100 ng/ml, Miltenyi Biotec) was added to the medium from day 9 until day 16. Cells were split 1:1 at day 11 and medium was switched to Neurobasal supplemented with 0.5× N2, 0.5× B27 without vitamin A, Penstrep, GlutaMAX. At day 16, cells were split 1:2 and the medium was switched to terminal differentiation medium consisting of Neurobasal-A supplemented with 1× B27 supplement without vitamin A, GlutaMAX, Penstrep containing brain-derived neurotrophic factor (BDNF, 10 ng/ml; Miltenyi Biotec), ascorbic acid (0.2 mM, Sigma), glial cell line-derived neurotrophic factor (GDNF, 10 ng/ml; Miltenyi Biotec), dibutyryl cAMP (0.2 mM; Sigma), SR11237 (100 nM, Tocris), and DAPT (10 µM; Tocris). On day 20, ventral midbrain neural progenitors were cryopreserved and quality controlled. To obtain mature neurons, neural progenitor cells were terminally differentiated on coverslips previously coated with poly-D-lysine (50 µg/ml, Life Technologies) and mouse laminin (1 µg/ml, Sigma) in terminal differentiation medium for an additional 40 days.

## Characterization of hiPSCs, progenitor cells, and mature neurons

For the characterization of hiPSCs, ventral midbrain floor plate progenitor cells and mature DAN were fixed for 15 min in 4% formaldehyde at the pluripotency stage, day 20 of differentiation or day 60 of

differentiation, respectively. Cells were blocked for 1 hr at room temperature with 3% normal goat serum+0.3% Triton X-100 (Sigma) in DPBS supplemented with $Ca^{2+}$ and $Mg^{2+}$ (Life Technologies). Primary antibodies were incubated overnight at 4°C in blocking solution. Secondary antibodies were incubated for 1 hr at room temperature in blocking solution. Coverslips were mounted in Mowiol (Sigma) and imaged on an upright Nikon A1R confocal microscope equipped with a DIC N2 ×20 lens (NA 0.75). Z-stacks were acquired with pinhole of 1 Airy unit, a Galvano scanner with line averaging of 2, image size of 1024×1024 pixels, and step intervals of 2 μm.

The following antibodies were used: mouse IgG1 anti-SOX2 (1:200 [Santa Cruz]), rabbit anti-OCT4 (1:50 [Abcam]), mouse IgG1 anti-NANOG (1:50 [Santa Cruz]), mouse IgM anti-TRA-1–81 (1:100 [Sigma]), rabbit anti-LMX1A/B (1:1000 [Millipore]), mouse IgG2a anti-FOXA2 (1:250 [Santa Cruz]), mouse IgG1 anti-OTX2 (1:100 [Santa Cruz]), rabbit anti-TH (1:500 [Sigma]), mouse IgG1 anti-MAP2 (1:1000 [Sigma]), Alexa Fluor-488/Alexa Fluor-555/Alexa Fluor-647 conjugated secondary antibodies (1:500 [Invitrogen]).

## In vitro ca²⁺ imaging and drug treatment

Day 40 DAN were treated for the following 19 days with 1 μM nicotine or with 1 μM nicotine plus 1 μM of the nicotinic receptor antagonist mecamylamine hydrochloride (Sigma). At day 50, DAN were transduced with a lentiviral vector expressing mRuby2-T2A-GCaMP6f (Addgene plasmid # 197595). Transduction efficacy was assessed by directly visualizing mRuby2 red fluorescence. Drug treatment was interrupted 1 day before analysis. For calcium imaging, cell culture medium was switched to Brain-Phys Imaging medium (STEMCELL Technologies) supplemented with B27 plus (Life Technologies), PenStrep (Life Technologies), GDNF (10 ng/ml, Miltenyi Biotec), BDNF (10 ng/ml, Miltenyi Biotec), dibutyryl cAMP (0.2 mM, Sigma), and SR11237 (100 nM, Tocris).

Cells were imaged at 37°C at a wide-field fluorescence microscope (Nikon) equipped with a Hamamatsu camera and a ×20/NA = 0.9 water-immersion objective. For each plate, at least two areas were imaged for 6 min at a frame rate of 5 Hz. Images were analyzed using Fiji (*Schindelin et al., 2012*). All visible cell bodies in the field of view defined an ROI, background fluorescence was subtracted to correct for bleaching. Basal fluorescence $F_0$ in each ROI was determined by the minimal average of five frames and the change of fluorescence over time was calculated by $F-F_0/F_0$. Values above 2×StdDev of the baseline frames were considered as activity. Activity is expressed as active frames/total measured frames. The maximal amplitude was counted as amplitude for each neuron.

## Quantification and statistical analysis

GraphPad Prism was used for visualization and statistical analysis of results from all but the sequencing experiments. Male and female flies were quantified separately, and results pooled later because they did not show significant differences. Datasets were checked for normal distribution and then analyzed with ANOVA and subsequent multiple pairwise comparisons using Dunnett's test when comparing to a general control, or Bonferroni correction of p-values when comparing different control-test pairs.

## Resource availability

### Lead contact

For further information and requests for resources and reagents, contact Patrik Verstreken (patrik.verstreken@kuleuven.be).

### Material availability

Plasmids and fly lines generated in this study are available on request.

## Acknowledgements

We thank Ann Geens, Eren Can Eksi, Vinoy Vijayan, Liesbeth Deaulmerie, Willem van den Bergh, Marianna Decet and Sara Aibar, Bart De Strooper, Joris de Wit, Roman Praschberger, the members of the Verstreken and Aerts labs and the VIB Bioimaging core for help and comments. We thank Atefeh Pooryasin and the Bloomington Drosophila Stock Center (NIH P40OD018537) for fly lines. Human tissue was kindly provided by the Parkinson's UK Brain Bank at Imperial College London, funded by Parkinson's UK, a charity registered in England and Wales (258197) and in Scotland (SC037554).

Research support was provided by ERC CoGs (PV and SA) and ERC AdG (PV), the Research Foundation Flanders (FWO), a Methusalem grant of the Flemish government, IMI2, Opening the Future (KU Leuven fund), and VIB. UP was supported by a fellowship from DFG; CCA was supported by the Marie Skłodowska-Curie Actions Seal of Excellence from FWO (Ref: 204990/12ZY321N), ATB was supported by an EMBO long-term fellowship and JJ and SV were supported by a fellowship from FWO. PV is an alumnus of the FENS Kavli Network of Excellence.

## Additional information

### Competing interests

Patrik Verstreken: Reviewing editor, eLife. The other authors declare that no competing interests exist.

### Funding

| Funder | Grant reference number | Author |
| --- | --- | --- |
| European Research Council | CoG | Stein Aerts<br>Patrik Verstreken |
| European Research Council | AdG | Patrik Verstreken |
| Fonds Wetenschappelijk Onderzoek | | Jasper Janssens<br>Carles Calatayud Aristoy<br>Stein Aerts<br>Patrik Verstreken<br>Sven Vilain |
| Vlaamse Overheid | Methusalem | Patrik Verstreken |
| EU Innovative Medicines Initiative 2 | IMI2 | Patrik Verstreken |
| KU Leuven | Opening the future | Patrik Verstreken |
| Vlaams Instituut voor Biotechnologie | | Stein Aerts<br>Patrik Verstreken |
| Deutsche Forschungsgemeinschaft | | Ulrike Pech |
| European Molecular Biology Organization | | Adekunle T Bademosi |
| FENS-Kavli Network of Excellence | | Patrik Verstreken |
| Marie Skłodowska-Curie Actions Seal of Excellence | 204990/12ZY321N | Carles Calatayud Aristoy |

The funders had no role in study design, data collection and interpretation, or the decision to submit the work for publication.

### Author contributions

Ulrike Pech, Conceptualization, Data curation, Formal analysis, Funding acquisition, Investigation, Visualization, Methodology, Writing – original draft, Writing – review and editing; Jasper Janssens, Data curation, Formal analysis, Investigation, Visualization, Methodology, Writing – review and editing; Nils Schoovaerts, Sabine Kuenen, Sandra F Gallego, Samira Makhzami, Gert J Hulselmans, Suresh Poovathingal, Kristofer Davie, Jef Swerts, Methodology, Writing – review and editing; Carles Calatayud Aristoy, Formal analysis, Investigation, Methodology, Writing – review and editing; Adekunle T Bademosi, Funding acquisition, Investigation, Methodology, Writing – review and editing; Sven Vilain, Conceptualization, Funding acquisition, Methodology, Writing – review and editing; Stein Aerts, Supervision, Funding acquisition, Writing – review and editing; Patrik Verstreken, Conceptualization, Resources, Supervision, Funding acquisition, Visualization, Writing – original draft, Project administration, Writing – review and editing

## Author ORCIDs
Sabine Kuenen ![ORCID] https://orcid.org/0000-0001-9135-5293
Stein Aerts ![ORCID] https://orcid.org/0000-0002-8006-0315
Patrik Verstreken ![ORCID] https://orcid.org/0000-0002-5073-5393

## Ethics
Usage of postmortem brain samples was ethically approved (EC reference NH019 2019-02-01). Donors agreed to donate brain tissue via a donor consent form.

Reviewer #1 (Public review): https://doi.org/10.7554/eLife.98348.3.sa1
Reviewer #3 (Public review): https://doi.org/10.7554/eLife.98348.3.sa2
Author response https://doi.org/10.7554/eLife.98348.3.sa3

---

# Additional files

## Supplementary files
Supplementary file 1. List of gRNAs, gBlocks, and oligos used to generate and validate the fly lines and hiPSC line generated in this study. List of full fly genotypes and hiPSC lines used in this study.

Supplementary file 2. Full results of the Cell Ranger summaries (10x Genomics).

Supplementary file 3. Summary of the number of cells per cell cluster for each mutant and controls, including the raw number of cells, the percentage of cells for each cluster per mutant, and the percentage of cells for each mutant and controls per cluster.

Supplementary file 4. Summary of the parameters used to model differentially expressed gene (DEG)-cell number correlation.

Supplementary file 5. List of cell types and respective z-residuals (deviation from model) to determine affected cell types and those that are outside of the 95% confidence interval.

Supplementary file 6. List of the clusters with significant changes (based on cell number-adjusted differentially expressed genes [DEGs]) for each model, black cell clusters (cells with significant number of DEG) are listed as TRUE.

Supplementary file 7. Summarized results of the differentially expressed gene (DEG) analysis in fly model brains and postmortem human samples and the summary of the SynGO analysis.

MDAR checklist

## Data availability
The datasets generated in this study are publicly available on GEO (GSE235332), with separate subseries for human data (GSE235330) and *Drosophila* data (GSE228843). All data generated or analyzed during this study are included in the manuscript and supporting files.

The following datasets were generated:

| Author(s) | Year | Dataset title | Dataset URL | Database and Identifier |
|---|---|---|---|---|
| Janssens J, Pech U, Aerts S, Verstreken P | 2025 | Rescuing early Parkinson-induced hyposmia prevents dopaminergic system failure [fruit fly] | https://www.ncbi.nlm.nih.gov/geo/query/acc.cgi?acc=GSE228843 | NCBI Gene Expression Omnibus, GSE228843 |
| Janssens J, Pech U, Aerts S, Verstreken P | 2025 | Rescuing early Parkinson-induced hyposmia prevents dopaminergic system failure [human] | https://www.ncbi.nlm.nih.gov/geo/query/acc.cgi?acc=GSE235330 | NCBI Gene Expression Omnibus, GSE235330 |
| Janssens J, Pech U, Aerts S, Verstreken P | 2025 | Rescuing early Parkinson-induced hyposmia prevents dopaminergic system failure (SuperSeries) | https://www.ncbi.nlm.nih.gov/geo/query/acc.cgi?acc=GSE235332 | NCBI Gene Expression Omnibus, GSE235332 |

The following previously published dataset was used:

| Author(s) | Year | Dataset title | Dataset URL | Database and Identifier |
|---|---|---|---|---|
| Davie K, Janssens J, Koldere D, Aerts S | 2018 | A single-cell transcriptome atlas of the ageing *Drosophila* brain | https://www.ncbi.nlm.nih.gov/geo/query/acc.cgi?acc=GSE107451 | NCBI Gene Expression Omnibus, GSE107451 |

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

# Appendix 1

## Appendix 1—key resources table

| Reagent type (species) or resource | Designation | Source or reference | Identifiers | Additional information |
|---|---|---|---|---|
| Gene (*D. melanogaster*) | Lrrk | FlyBase | FBgn0038816 | |
| Gene (*D. melanogaster*) | Pink1 | FlyBase | FBgn0029891 | |
| Gene (*D. melanogaster*) | Rab39 | FlyBase | FBgn0029959 | |
| Gene (*D. melanogaster*) | Aux | FlyBase | FBgn0037218 | |
| Gene (*D. melanogaster*) | Synj | FlyBase | FBgn0034691 | |
| Strain, strain background (*Escherichia coli*) | One Shot top10 cells | Thermo Fisher Scientific | C404010 | Competent cells |
| Genetic reagent (*D. melanogaster*) | GH146-Gal4 (OPN-Gal4) | *Stocker et al., 1997*; Bloomington Drosophila Stock Center | BDSC_30026; FlyBase: FBti0016783 | |
| Genetic reagent (*D. melanogaster*) | GMR31F10-Gal4 (T1-Gal4) | *Jenett et al., 2012*; Bloomington Drosophila Stock Center | BDSC_49685; FlyBase: FBti0134817 | |
| Genetic reagent (*D. melanogaster*) | GMR58E02-Gal4 (PAM-Gal4) | *Jenett et al., 2012*; Bloomington Drosophila Stock Center | BDSC_41347; FlyBase: FBti0137105 | |
| Genetic reagent (*D. melanogaster*) | UAS-GCaMP3 | *Tian et al., 2009*; Bloomington Drosophila Stock Center | BDSC_32236; BDSC_32116 | |
| Genetic reagent (*D. melanogaster*) | UAS-hLRRK2 | *Venderova et al., 2009* | | |
| Genetic reagent (*D. melanogaster*) | UAS-hPINK1 | *Yang et al., 2006* | | |
| Genetic reagent (*D. melanogaster*) | UAS-YFP-Rab39 | *Zhang et al., 2007*; Bloomington Drosophila Stock Center | BDSC_9825; FlyBase: FBti0100878 | |
| Genetic reagent (*D. melanogaster*) | UAS-Synj | *Vanhauwaert et al., 2017* | | |
| Genetic reagent (*D. melanogaster*) | UAS-Aux | *Jacquemyn et al., 2023* | | |
| Genetic reagent (*D. melanogaster*) | LRRK[k.o.] | This study (see *Figure 1a* and the Materials and methods section) | | Knock-out fly line carrying an attP-flanked mini-white gene replacing the first common exon in all transcripts of lrrk |
| Genetic reagent (*D. melanogaster*) | hLRRK2 | This study (see *Figure 1a* and the Materials and methods section) | | Knock-in fly line carrying the wild-type human LRRK2 CDS at the lrrk locus |
| Genetic reagent (*D. melanogaster*) | hLRRK2[G2019S] | This study (see *Figure 1a* and the Materials and methods section) | | Knock-in fly line carrying the pathogenic human LRRK2[G2019S] CDS at the lrrk locus |
| Genetic reagent (*D. melanogaster*) | hLRRK2[Y1699C] | This study (see *Figure 1a* and the Materials and methods section) | | Knock-in fly line carrying the pathogenic human LRRK2[Y1699C] CDS at the lrrk locus |
| Genetic reagent (*D. melanogaster*) | PINK1[k.o.] | This study (see *Figure 1a* and the Materials and methods section) | | Knock-out fly line carrying an attP-flanked mini-white gene replacing the first common exon in all transcripts of pink1 |
| Genetic reagent (*D. melanogaster*) | hPINK1[P399L] | This study (see *Figure 1a* and the Materials and methods section) | | Knock-in fly line carrying the pathogenic human PINK1[P399L] CDS at the pink1 locus |
| Genetic reagent (*D. melanogaster*) | hPINK1[L347P] | This study (see *Figure 1a* and the Materials and methods section) | | Knock-in fly line carrying the pathogenic human PINK1[L347P] CDS at the pink1 locus |

*Appendix 1 Continued on next page*

*Appendix 1 Continued*

| Reagent type (species) or resource | Designation | Source or reference | Identifiers | Additional information |
|---|---|---|---|---|
| Genetic reagent (*D. melanogaster*) | Rab39$^{k.o.}$ | This study (see *Figure 1a* and the Materials and methods section) | | Knock-out fly line carrying an attP-flanked mini-white gene replacing the first common exon in all transcripts of rab39 |
| Genetic reagent (*D. melanogaster*) | hRab39B | This study (see *Figure 1a* and the Materials and methods section) | | Knock-in fly line carrying the wild-type human Rab39B CDS at the rab39 locus |
| Genetic reagent (*D. melanogaster*) | hRab39B$^{G192R}$ | This study (see *Figure 1a* and the Materials and methods section) | | Knock-in fly line carrying the pathogenic human Rab39B$^{G192R}$ CDS at the rab39 locus |
| Genetic reagent (*D. melanogaster*) | hRab39B$^{T168K}$ | This study (see *Figure 1a* and the Materials and methods section) | | Knock-in fly line carrying the pathogenic human Rab39B$^{T168K}$ CDS at the rab39 locus |
| Genetic reagent (*D. melanogaster*) | Aux$^{k.o.}$ | This study (see *Figure 1a* and the Materials and methods section) | | Knock-out fly line carrying an attP-flanked mini-white gene replacing the first common exon in all transcripts of aux |
| Genetic reagent (*D. melanogaster*) | Aux | This study (see *Figure 1a* and the Materials and methods section) | | Knock-in fly line carrying the wild-type *Drosophila* Aux CDS at the aux locus |
| Genetic reagent (*D. melanogaster*) | Aux$^{R927G}$ | This study (see *Figure 1a* and the Materials and methods section) | | Knock-in fly line carrying the pathogenic *Drosophila* Aux$^{R927G}$ CDS at the aux locus |
| Genetic reagent (*D. melanogaster*) | Synj$^{k.o.}$ | This study (see *Figure 1a* and the Materials and methods section) | | Knock-out fly line carrying an attP-flanked mini-white gene replacing the first common exon in all transcripts of synj |
| Genetic reagent (*D. melanogaster*) | Synj | This study (see *Figure 1a* and the Materials and methods section) | | Knock-in fly line carrying the wild-type *Drosophila* Synj CDS at the synj locus |
| Genetic reagent (*D. melanogaster*) | Synj$^{R258Q}$ | This study (see *Figure 1a* and the Materials and methods section) | | Knock-in fly line carrying the pathogenic *Drosophila* Synj$^{R258Q}$ CDS at the synj locus |
| Cell line (*H. sapiens*) | KOLF2-1J wild-type (LRRK2$^{WT}$) control iPSCs | *Pantazis et al., 2022*; from the Jackson Laboratory under the iPSC Neurodegenerative Disease Initiative | Product code: JIPSC001000 | |
| Cell line (*H. sapiens*) | KOLF2-1J LRRK2$^{G2019S/G2019S}$ (LRRK2$^{G2019S}$) iPSCs | This study (see *Figure 6—figure supplement 1* and the Materials and methods section) | | Knock-in KOLF2-1J iPSC line carrying the pathogenic p.G2019S mutation in the LRRK2 locus |
| Transfected construct (*H. sapiens*) | pLenti-hSyn1-mRuby2-T2A-GCaMP6f | This study (see the Materials and methods section) | RRID:Addgene_197595 | Lentiviral transfer vector for the neuronal expression of mRuby2-T2A-GCaMP6f (fluorescent reporter for calcium imaging) |
| Biological sample (*H. sapiens*) | Postmortem brain tissue | Parkinson's UK Brain Bank at the Imperial College London | EC reference NH019 2019-02-01 | |
| Antibody | Mouse monoclonal IgG1 anti-SOX2 | Santa Cruz | sc-365823 | IF (1:200) |
| Antibody | Rabbit polyclonal anti-OCT4 | Abcam | ab19857 | IF (1:50) |
| Antibody | Mouse monoclonal IgG1 anti-NANOG | Santa Cruz | sc-293121 | IF (1:50) |
| Antibody | Mouse monoclonal IgM anti-TRA-1–81 | Sigma | MAB4381 | IF (1:100) |
| Antibody | Rabbit polyclonal anti-LMX1A/B | Millipore | AB10533 | IF (1:1000) |
| Antibody | Mouse monoclonal IgG2a anti-FOXA2/HNF-3β | Santa Cruz | sc-101060 | IF (1:250) |

*Appendix 1 Continued*

| Reagent type (species) or resource | Designation | Source or reference | Identifiers | Additional information |
|---|---|---|---|---|
| Antibody | Mouse monoclonal IgG1k anti-OTX2 | Santa Cruz | sc-514195 | IF (1:100) |
| Antibody | Rabbit polyclonal anti-TH | Sigma | AB152 | IF (1:500) |
| Antibody | Mouse monoclonal IgG1 anti-MAP2 | Sigma | M1406 | IF (1:1000) |
| Antibody | Goat polyclonal Alexa Fluor-488 conjugated secondary antibody | Invitrogen | Anti-rabbit: # A-11034 for OCT4/TH Anti-mouse IgG1: # A-21121 for Nanog/OTX2 | IF (1:500) |
| Antibody | Goat polyclonal Alexa Fluor-555 conjugated secondary antibody | Invitrogen | Anti-mouse IgG1: # A-21127 for SOX2 Anti-mouse IgG2a: # A-21137 for FOXA2 | IF (1:500) |
| Antibody | Donkey/Goat polyclonal Alexa Fluor-647 conjugated secondary antibody | Invitrogen | Anti-rabbit: # A-31573 for LMX1A Anti-mouse IgM: # A-21238 for TRA-1–81 Anti-mouse IgG1: # A-21240 for MAP2 | IF (1:500) |
| Recombinant DNA reagent | pCFD4: U6:1-gRNA U6:3-gRNA | *Port et al., 2014* | RRID:Addgene_49411 | |
| Recombinant DNA reagent | pUC19 | *Norrander et al., 1983* | RRID:Addgene_50005 | |
| Recombinant DNA reagent | pUAST.attB | *Bischof et al., 2007* | GenBank:EF362409.1 | |
| Recombinant DNA reagent | pFL44S{w+}-attB | *Khuong et al., 2013* | | |
| Recombinant DNA reagent | pWhite-STAR | *Choi et al., 2009* | | |
| Recombinant DNA reagent | 2XMyc-LRRK2-WT | *Greggio et al., 2008* | RRID:Addgene_25361 | |
| Recombinant DNA reagent | pLenti6-DEST PINK1-V5 WT | *Beilina et al., 2005* | RRID:Addgene_13320 | |
| Sequence-based reagent | Primers, gRNAs, oligos, gBlocks, ssODN | Integrated DNA Technologies (IDT) | | See *Supplementary file 1* |
| Sequence-based reagent | iPSC gRNA | Synthego | | See *Supplementary file 1* |
| Peptide, recombinant protein | Cas9 HiFi protein | Integrated DNA Technologies (IDT) | Alt-R S.p. HiFi Cas9 Nuclease V3, 500 µg; #1081061 | |
| Peptide, recombinant protein | Dispase | Sigma-Aldrich | D4818 | |
| Peptide, recombinant protein | Collagenase | Invitrogen, CA, USA | 17100-017 | |
| Chemical compound, drug | Ringer's solution | *Estes et al., 1996* | | |
| Chemical compound, drug | TMRE | Abcam, Cambridge, UK | ab113852 | |
| Chemical compound, drug | Trypsin-EDTA | Invitrogen, CA, USA | 25300054 | |
| Chemical compound, drug | Nicotine | Sigma-Aldrich | N3876 | |
| Commercial assay, kit | NEBuilder HiFi DNA Assembly | New England Biolabs, Inc (NEB) | NEB #E2621 | |
| Commercial assay, kit | Maxwell RSC instrument and kit | Promega | AS8500, AS1340 | |
| Commercial assay, kit | Chromium Single Cell 3′ Library & Gel Bead Kit v2 | 10x Genomics | PN-120237 | |
| Commercial assay, kit | Chromium Single Cell A Chip Kit | 10x Genomics | PN-1000009 | |

*Appendix 1 Continued on next page*

*Appendix 1 Continued*

| Reagent type (species) or resource | Designation | Source or reference | Identifiers | Additional information |
|---|---|---|---|---|
| Commercial assay, kit | Chromium i7 Multiplex Kit | 10x Genomics | PN-120262 | |
| Commercial assay, kit | Chromium Next GEM Single Cell 3' Kit v3 | 10x Genomics | PN-1000075 | |
| Commercial assay, kit | Chromium Next GEM Chip B Single Cell Kit | 10x Genomics | PN-1000073 | |
| Commercial assay, kit | Single Index Kit T Set A | 10x Genomics | PN-1000213 | |
| Software, algorithm | Fiji | *Schindelin et al., 2012* | RRID:SCR_002285 | https://imagej.net/Fiji |
| Software, algorithm | Cell Ranger (v 3.0.2) | 10x Genomics | RRID:SCR_017344 | https://www.10xgenomics.com/support/software/cell-ranger/latest |
| Software, algorithm | GOrilla | *Eden et al., 2009* | RRID:SCR_006848 | http://cbl-gorilla.cs.technion.ac.il/ |
| Software, algorithm | GraphPad Prism | GraphPad software, CA, USA | RRID:SCR_002798 | https://www.graphpad.com/scientific-software/prism/ |
| Software, algorithm | Vib-singlecell-nf/vsn-pipelines (v 0.27.0) | *Flerin et al., 2021* | | https://github.com/vib-singlecell-nf/vsn-pipelines |
| Software, algorithm | Python (v 3.7.3) | Python | RRID:SCR_008394 | http://www.python.org/ |
| Software, algorithm | pandas (v 0.25.1) | GitHub | RRID:SCR_018214 | https://github.com/pandas-dev/pandas |
| Software, algorithm | numpy (v 1.19.1) | *Harris et al., 2020* | RRID:SCR_008633 | http://www.numpy.org/ |
| Software, algorithm | matplotlib (v 3.3.0) | GitHub | RRID:SCR_008624 | https://github.com/matplotlib/matplotlib |
| Software, algorithm | seaborn (v 0.9.0) | GitHub | RRID:SCR_018132 | https://github.com/mwaskom/seaborn |
| Software, algorithm | statsmodels (v 0.10.1) | *Seabold and Perktold, 2010* | RRID:SCR_016074 | http://www.statsmodels.org/ |
| Software, algorithm | scipy (v 1.4.1) | *Virtanen et al., 2020* | RRID:SCR_008058 | http://www.scipy.org/ |
| Software, algorithm | adjustText (v 0.7.3) | GitHub | RRID:SCR_022260 | https://github.com/Phlya/adjustText |
| Software, algorithm | loompy (v 3.0.6) | GitHub | RRID:SCR_016666 | https://github.com/linnarsson-lab/loompy |
| Software, algorithm | sklearn (v 0.20.1) | *Pedregosa et al., 2011* | RRID:SCR_019053 | https://scikit-learn.org/stable/modules/generated/sklearn.decomposition.NMF.html |
| Software, algorithm | patsy (v 0.5.1) | *Smith, 2018* | | https://github.com/pydata/patsy |
| Software, algorithm | venn (v 0.1.3) | Python | | https://pypi.org/project/venn/ |
| Software, algorithm | R (v 3.6.1) | R Foundation | RRID:SCR_001905 | http://www.r-project.org/ |
| Software, algorithm | MAST (v 1.12.0) | GitHub | RRID:SCR_016340 | https://github.com/RGLab/MAST/ |
| Software, algorithm | SCopeLoomR (v 0.5.1) | *De Waegeneer et al., 2022* | | https://github.com/aertslab/SCopeLoomR |
| Other | SCope Resource Website | *Davie et al., 2018* | | Visualization tool for large-scale sc-RNA-seq datasets; http://scope.aertslab.org |

