## [Editor Report · eLife Assessment]

This is an **important** study demonstrating that anosmia in Parkinson's disease patients is due to dysfunction in cholinergic neurons. This study provides **compelling** evidence, using scRNA sequencing, that cholinergic olfactory projection neurons (OPN) are consistently affected in five different fruit fly models of Parkinson's disease, exhibiting synaptic dysfunction before the onset of motor deficits. Comparisons with scRNA sequencing of patients' human brain samples reveals similar synaptic gene deregulation in cholinergic neurons of patients. This study points the possibility that targeting cholinergic neurons could be a potential avenue for early diagnosis and intervention in PD.

---

## [Referee Report · Reviewer #1 (Public review)]

In Pech et al. the authors take advantage of a genetic model organism to investigate the convergent impact of multiple mutations linked to Parkinson's Disease (PD). To investigate this question they leverage *Drosophila* genetics to create wild type and mutant alleles for five different mutations linked to PD. An additional novel focus of this work is an examination of the animals in an early phase before apparent dopaminergic degeneration. Having generated this resource, authors discover apply an impressive array of experiments including behavioural assays, calcium imaging and single-cell profiling. They also cross-validate their findings in human PD brains. Strikingly, the authors discover common dysregulated genes between fly and human that converges on synaptic dysregulation. Finally, they demonstrate that even in early timepoints, there is extensive dysfunction of olfactory projection neuron calcium.

This is a fantastic, comprehensive, timely and landmark pan-species work that demonstrates the convergence of multiple familial PD mutations onto a synaptic program. It is extremely well written and the authors have addressed all my comments in this review. I recommend this work be published as soon as possible.

---

## [Referee Report · Reviewer #3 (Public review)]

Summary:

This study investigates the cellular and molecular events leading to hyposmia, an early dysfunction in Parkinson's disease (PD), which develops up to 10 years prior to motor symptoms. The authors use five *Drosophila* knock-in models of familial PD genes (LRRK2, RAB39B, PINK1, DNAJC6 (Aux), and SYNJ1 (Synj)), three expressing human genes and two Drosophila genes with equivalent mutations.

The authors carry out single-cell RNA sequencing of young fly brains and single-nucleus RNA sequencing of human brain samples. The authors found that cholinergic olfactory projection neurons (OPN) were consistently affected across the fly models, showing synaptic dysfunction before the onset of motor deficits, known to be associated with dopaminergic neuron (DAN) dysfunction.

Single-cell RNA sequencing revealed significant transcriptional deregulation of synaptic genes in OPNs across all five fly PD models. This synaptic dysfunction was confirmed by impaired calcium signalling and morphological changes in synaptic OPN terminals. Furthermore, these young PD flies exhibited olfactory behavioural deficits that were rescued by selective expression of wild-type genes in OPNs.

Single-nucleus RNA sequencing of post-mortem brain samples from PD patients with LRRK2 risk mutations revealed similar synaptic gene deregulation in cholinergic neurons, particularly in the nucleus basalis of Meynert (NBM). Gene ontology analysis highlighted enrichment for processes related to presynaptic function, protein homeostasis, RNA regulation, and mitochondrial function.

This study provides compelling evidence for the early and primary involvement of cholinergic dysfunction in PD pathogenesis, preceding the canonical DAN degeneration. The convergence of familial PD mutations on synaptic dysfunction in cholinergic projection neurons suggests a common mechanism contributing to early non-motor symptoms like hyposmia. The authors also emphasise the potential of targeting cholinergic neurons for early diagnosis and intervention in PD.

Strengths:

This study presents a novel approach, combining multiple mutants to identify salient disease mechanisms. The quality of the data and analysis is of a high standard, providing compelling evidence for the role of OPN neurons in olfactory dysfunction in PD. The authors also provide evidence to show that early olfactory defects lead to later dopaminergic neuron dysfunction. The comprehensive single-cell RNA sequencing data from both flies and humans is a valuable resource for the research community. The identification of consistent impairments in cholinergic olfactory neurons, at early disease stages, is a powerful finding that highlights the convergent nature of PD progression. The comparison between fly models and human patients' brains provides strong evidence of the conservation of molecular mechanisms of disease, which can be built upon in further studies using flies to prove causal relationships between the defects described here and neurodegeneration.

The identification of specific neurons involved in olfactory dysfunction opens up potential avenues for diagnostic and therapeutic interventions.

---

## [Author Response]

The following is the authors’ response to the original reviews.

We thank the reviewers for their comments and provide answers /clarifications and new data; There were 3 important recurrent points we already address here:

(a) The reviewers were concerned that the observed motor defects (measured by startle induced negative geotaxis- “SING”) where a reasonable behavioral measure of DAN function.

Previously, [69] already linked synaptic loss of the dopaminergic PAM neurons to SING impairments. Furthermore, in a separate paper that we recently posted on BioRxiv, we show that the SING defects in PD mutants are rescued when the flies are fed L-DOPA (42). In this same paper we also show a very strong correlation between SING defects and defects in dopaminergic synaptic innervation of PAM DAN onto Mushroom body neurons. Both experiments suggest that the motor defects are the result of defects in dopamine release. Altogether, these data suggest that the combination of the SING assay and a quantification of the synaptic region of PAM DAN onto Mushroom body neurons is a suitable measure for DAN function.

(b) The reviewers asked if the OPN dysfunction in young animals is connected to dopaminergic neuron (DAN) dysfunction in later life;

We have conducted additional experiments and have included the results (new Figure 6): Our young PD mutants (we included Aux^R927G^, Synj^R258Q^ and LRRK2^G2019S^) show olfactory defects, but normal DAN function (measured by assessing the TH-labeled synaptic area onto the Mushroom body neurons and by SING). Aged PD mutants show both olfactory defects and DAN dysfunction. When we express the wildtype PD gene in (a.o.) OPN of PD mutants using the GH146-Gal4 (that does not drive expression in DAN) we are able to rescue the DAN defects (synaptic area and SING) that occur later in life. This indeed suggests there is a cell non-autonomous positive effect on DAN dysfunction that occurs at later stages in the life of our PD mutants (new Figure 6a).

In a set of independent experiments, we also fed one of our mutants (LRRK2^G2019S^) nicotine, activating Nicotinic acetylcholine receptors (that are also activated by the release of acetylcholine from cholinergic neurons such as OPN). While nicotine does not rescue the olfactory preference defect, the OPN synapse morphology defect or the OPN-associated defects in Ca^2+^-imaging in LRRK2^G2019S^ mutants (Figure 6b), it does rescue the DAN-associated defects, including SING, synapse loss and defects in Ca^2+^-imaging (Figure 6c).

Finally, we generated human induced dopaminergic neurons derived from iPSC with a LRRK2^G2019S^ mutation and incubated these neurons with nicotine. Again, this induced a rescue of a LRRK2-mutant-induced defect in neuronal activity measured by Ca^2+^-imaging. This is specific to nicotine since the rescue was absent when cells were also incubated with mecamylamine, a non-competitive antagonist of nicotinic acetylcholine receptors, trumping the effects of nicotine (Figure 6d-e").

(c) The reviewers indicated that the GH146 Gal 4 driver is expressed in other cells than OPN and thus, they noted that the defects we observe may not only be the result of OPN dysfunction.

It is correct that GH146-dependent Gal expression includes OPNs (that are cholinergic) and one pair of inhibitory APL neurons (that are GABAergic) (51, 53). We have adapted the text to explicitly state this. There are only 2 APL per fly brain and our single cell sequencing experiment does not have the resolution to allow us to test if these neurons had a significant number of DEG. However, as indicated above (in (b)), we are able to rescue DAN dysfunction by mimicking cholinergic output (application of nicotine). These data do not exclude that APL-neuron problems contribute to the defects we observe in our PD mutants, but they do suggest that cholinergic output is critical to maintain normal DAN function.

**Public Reviews:**

**Reviewer #1 (Public Review):**
This is a fantastic, comprehensive, timely, and landmark pan-species work that demonstrates the convergence of multiple familial PD mutations onto a synaptic program. It is extremely well written and I have only a few comments that do not require additional data collection.

Thank you for this enthusiastic endorsement.

Major Comments:neurons and the olfactory system are acutely impacted by these PD mutations. However, I wonder if this is the case:(1) In the functional experiments performing calcium imaging on projection neurons I could not find a count of cell bodies across conditions. Since the loss of OPNs could explain the reduced calcium signal, this is a critical control to perform. A differential abundance test on the single-cell data would also suffice here and be easy for the authors to perform with their existing data.

This is indeed an important number, and we had included this in the Supplemental figure 2a.

Also, the number of DAN and Visual projection neurons were not significantly different between the genotypes (Supplemental Figure 2a in the manuscript).

(2) One of the authors' conclusions is that cholinergica. Most *Drosophila* excitatory neurons are cholinergicand only a subpopulation appear to be dysregulated by these mutations. The authors point out that visual neurons also have many DEGs, couldn't the visual system also be dysregulated in these flies? Is there something special about these cholinergic neurons versus other cholinergic neurons in the fly brain? I wonder if they can leverage their nice dataset to say something about vulnerability.

Yes, the reviewer is right, and we have changed our wording to be more specific. The reviewer also noted correctly that neurons in the visual system rank high in terms of number of DEGs, but we did not conduct elaborate experiments to assess if these visual system neurons are functional. Of note, several of our mutants show (subtle) electroretinogram defects, that are a measure of visual system integrity, but further work is needed to determine the origin of these defects.

The question about the nature of the underlying vulnerability pathways is interesting. In preliminary work we have selected a number of DEGs common to vulnerable cells in several PD mutants, and conducted a screen where we manipulated the expression of these DEGs and looked for rescue of the olfactory preference defects in our PD mutants. The strongest genetic interaction was with genes encoding proteins involved in proteostasis (Atg8/LC3, Lamp1 and Hsc70-4) (Author response image 1). While interesting, these results require further work to understand the underlying molecular mechanisms. We present these preliminary data here but have not included them in the main manuscript.

b. As far as I can tell, the cross-species analysis of DEGs (Figure 3) is agnostic to neuronal cell type, although the conclusion seems to suggest only cholinergic neurons were contrasted. Is this correct? Could you please clarify this in the text as it's an important detail. If not, Have the authors tried comparing only cholinergic neuron DEGs across species? That would lend strength to their specificity argument. The results for the NBM are impressive. Could the authors add more detail to the main text here about other regions to the main text?

The reviewer is correct that we compiled the DEG of all affected cells, the majority of which are cholinergic neurons.

For the human data we focused on the NBM samples, because it contained the highest fraction of cholinergic neurons (as compared to the other 2 regions), but even so, it was not possible to analyze the cholinergic neurons alone because the fraction of cholinergic neurons in the human material was too low to be statistically analyzed independently. Note that both wildtype and PD samples contained a low number of cholinergic neurons (i.e. the DEG differences we detected were not the result of sequencing different types of cells - see also Supplemental Figure 3b and d). We have indicated this more clearly in the text.

c. Uniquely within the human data, are cholinergic neurons more dysregulated than others? I understand this is not an early timepoint but would still be useful to discuss.

As indicated in the previous point, unfortunately the fraction of cholinergic neurons in the human material was low and we were not able to analyze these cells on their own.

**Author response image 1. sa3fig1:** Upregulation of protein homeostasis rescues hyposmia across familial models of PD. Results of a behavioral screen for cell-specific rescue of olfactory preference defects of young PD fly models using up and downregulation of deregulated genes in affected cell types. Genes implicated in the indicated pathways are over expressed or knocked down using GH146-Gal4 (OPN>) and UAS-constructs (over expression or RNAi) . UAS-only (-) and OPN>UAS (+) were scored in parallel and are compared to each other. n.d. not determined; Bars represent mean ± s.e.m.; gray zone indicates the variance of controls; n≥5 independent experiments per genotype, with ~50 flies each; red bars: p<0.05 in ANOVA and Bonferroni-corrected comparison to UAS-only control.

d. In the discussion, the authors say that olfactory neurons are uniquely poised to be dysregulated as they are large and have high activity. Is this really true compared to other circuits? I didn't find the references convincing and I am not sure this has been borne out in electron microscopy reconstructions for anatomy.

We agree and have toned down this statement.

**Reviewer #2 (Public Review):**
Summary:Pech et al selected 5 Parkinson's disease-causing genes, and generated multiple*Drosophila* lines by replacing the Drosophila lrrk, rab39, auxilin (aux), synaptojanin(synj), and Pink1 genes with wild-type and pathogenic mutant human or *Drosophila* cDNA sequences. First, the authors performed a panel of assays to characterize the phenotypes of the models mentioned above. Next, by using single-cell RNA-seq and comparing fly data with human postmortem tissue data, the authors identified multiple cell clusters being commonly dysregulated in these models, highlighting the olfactory projection neurons. Next, by using selective expression of Ca^2+^-sensor GCaMP3 in the OPN, the authors confirmed the synaptic impairment in these models, which was further strengthened by olfactory performance defects.Strengths:The authors overall investigated the functionality of PD-related mutations at endogenous levels and found a very interesting shared pathway through singlecell analysis, more importantly, they performed nice follow-up work using multiple assays.Weaknesses:While the authors state this is a new collection of five familial PD knock-in models, the Aux^R927G^ model has been published and carefully characterized in Jacquemyn et al., 2023. ERG has been performed for Aux R927G in Jacquemyn et al., 2023, but the findings are different from what's shown in Figure 1b and Supplementary Figure 1d, which the authors should try to explain.

We should have explained this better: the ERG assay in Jacquemyn et al., and here, in Pech et al., are different. While the ERGs in our previous publication were recorded under normal endogenous conditions, the flies in our current study were exposed to constant light for 7 days. This is often done to accelerate the degeneration phenotype. We have now indicated this in the text (and also refer to the different experimental set up compared to Jacquemyn et al).

Moreover, according to the authors, the hPINK1control was the expression of human PINK1 with UAS-hPINK1 and nsyb-Gal4 due to technical obstacles. Having PINK1 WT being an overexpression model, makes it difficult to explain PINK1 mutant phenotypes. It will be strengthened if the authors use UAS-hPINK1 and nsyb-Gal4 (or maybe ubiquitous Gal4) to rescue hPink1L347P and hPink1P399L phenotypes.

The UAS-hPink1 was originally created by the Lu lab and has been amply used before in Pink1 loss-of-function backgrounds (e.g. in [99]). In our work, the control we refer to was UAS-hPink1 expression (driven by nSyb-gal4) in a Pink1 knock-out background. For unknown reasons we were unable to replace the fly Pink1 with a human pink1 cDNA, we explained this in the methods section and added a remark in the new manuscript.

In addition, although the authors picked these models targeting different biology/ pathways, however, Aux and Synj both act in related steps of Clathrin-mediated endocytosis, with LRRK2 being their accessory regulatory proteins. Therefore, is the data set more favorable in identifying synaptic-related defects?

We picked these particular mutants, as they were the first we created in the context of a much larger collection of “PD flies” (see also [42]). We have made adaptations to the text to tone down the statement on the broad selection of mutants.

GH146-GAL4+ PNs are derived from three neuroblast lineages, producing both cholinergic and GABAergic inhibitory PNs (Li et al, 2017). Therefore, OPN neurons have more than "cholinergic projection neurons". How do we know from singlecell data that cholinergic neurons were more vulnerable across 5 models?

The reviewer is correct that GH146 drives expression in other cells than OPN and we now clearly state this in the text. We do present additional arguments that substantiate our conclusion that cholinergic neurons are affected: (1) our single cell sequencing identifies the most DEGs in cholinergic neurons. (2) nicotine (a compound activating cholinergic receptors) rescues dopamine-related problems in old PD-mutant flies. (3) Likewise, nicotine also alleviates problems we observed in LRRK2 mutant human induced dopaminergic neurons and this is blocked by mecamylamine, a non-competitive antagonist of nicotinic acetylcholine receptors.

In Figure 1b, the authors assumed that locomotion defects were caused by dopaminergic neuron dysfunction. However, to better support it, the author should perform rescue experiments using dopaminergic neuron-specific Gal4 drivers. Otherwise, the authors may consider staining DA neurons and performing cell counting. Furthermore, the authors stated in the discussion, that "We now place cholinergic failure firmly ahead of dopaminergic system failure in flies", which feels rushed and insufficient to draw such a conclusion, especially given no experimental evidence was provided, particularly related to DA neuron dysfunction, in this manuscript.

Previously, [69] already linked synaptic loss of the dopaminergic PAM neurons to locomotion impairments (measured by SING). Furthermore, in a separate paper we show that the motor defects (SING) observed in PD mutants are rescued when the flies are fed L-DOPA, but not D-DOPA (42). In this same paper, we also show a significant correlation between SING defects and defects in dopaminergic synaptic innervation of PAM DAN onto Mushroom body neurons. We have referred to both articles in the revised manuscript.

The statement on cholinergic failure ahead of dopaminergic failure was made in the context of the sequence of events: young flies did not show DAN defects, but they did display olfactory defects. The statement was indeed not meant to imply causality. However, we have now conducted new experiments where we express wild type PD genes using GH146-Gal4 (that does not express in DAN) in the PD mutants and assess dopaminergic-relevant phenotypes later in life (see also new Figure 6 in the manuscript). This shows that GH146Gal4-specific rescue is sufficient to alleviate the DAN-dependent SING defects in old flies. Likewise, as indicated above, application of nicotine is also sufficient to rescue the DAN-associated defects (in PD mutant flies and human induced mutant dopaminergic neurons).

It is interesting to see that different familial PD mutations converge onto synapses. The authors have suggested that different mechanisms may be involved directly through regulating synaptic functions, or indirectly through mitochondria or transport. It will be improved if the authors extend their analysis on Figure 3, and better utilize their single-cell data to dissect the mechanisms. For example, for all the candidates listed in Figure 3C, are they all altered in the same direction across 5 models?

This is indeed the case: the criteria for "commonly deregulated" included that the DEGs are changed in the same direction across several mutants. We ranked genes according to their mean gene expression across the mutants as compared it to the wildtype control: i.e. only if the DEGs are all up- or all down-regulated they end up on the top or bottom of our list. We added a remark in the revised manuscript. In preliminary work we also selected a number of the DEGs and conducted a screen where we manipulated the expression of these genes looking for rescue of the olfactory preference defects in our PD mutants. The strongest genetic interaction was with genes encoding proteins involved in proteostasis (Atg8/LC3, Lamp1 and Hsc70-4; and we also show a genetic interaction between EndoA and Lrrk in this work and in Matta et al., 2012) (Author response image 1 above). While interesting, these results require further work to understand the underlying molecular mechanisms. We present these preliminary data here, but have not included them in the main manuscript.

While this approach is carefully performed, the authors should state in the discussions the strengths and the caveats of the current strategy. For example, what kind of knowledge have we gained by introducing these mutations at an endogenous locus? Are there any caveats of having scRNAseq at day 5 only but being compared with postmortem human disease tissue?

We have included a “strengths and caveats section” in the discussion addressing these points.

**Reviewer #3 (Public Review):**
Summary:This study investigates the cellular and molecular events leading to hyposmia, an early dysfunction in Parkinson's disease (PD), which develops up to 10 years prior to motor symptoms. The authors use five *Drosophila* knock-in models of familial PD genes (LRRK2, RAB39B, PINK1, DNAJC6 (Aux), and SYNJ1 (Synj)), three expressing human genes and two Drosophila genes with equivalent mutations.The authors carry out single-cell RNA sequencing of young fly brains and singlenucleus RNA sequencing of human brain samples. The authors found that cholinergic olfactory projection neurons (OPN) were consistently affected across the fly models, showing synaptic dysfunction before the onset of motor deficits, known to be associated with dopaminergic neuron (DAN) dysfunction.Single-cell RNA sequencing revealed significant transcriptional deregulation of synaptic genes in OPNs across all five fly PD models. This synaptic dysfunction was confirmed by impaired calcium signalling and morphological changes in synaptic OPN terminals. Furthermore, these young PD flies exhibited olfactory behavioural deficits that were rescued by selective expression of wild-type genes in OPNs.Single-nucleus RNA sequencing of post-mortem brain samples from PD patients with LRRK2 risk mutations revealed similar synaptic gene deregulation in cholinergic neurons, particularly in the nucleus basalis of Meynert (NBM). Gene ontology analysis highlighted enrichment for processes related to presynaptic function, protein homeostasis, RNA regulation, and mitochondrial function.This study provides compelling evidence for the early and primary involvement of cholinergic dysfunction in PD pathogenesis, preceding the canonical DAN degeneration. The convergence of familial PD mutations on synaptic dysfunction in cholinergic projection neurons suggests a common mechanism contributing to early non-motor symptoms like hyposmia. The authors also emphasise the potential of targeting cholinergic neurons for early diagnosis and intervention in PD.Strengths:This study presents a novel approach, combining multiple mutants to identify salient disease mechanisms. The quality of the data and analysis is of a high standard, providing compelling evidence for the role of OPN neurons in olfactory dysfunction in PD. The comprehensive single-cell RNA sequencing data from both flies and humans is a valuable resource for the research community. The identification of consistent impairments in cholinergic olfactory neurons, at early disease stages, is a powerful finding that highlights the convergent nature of PD progression. The comparison between fly models and human patients' brains provides strong evidence of the conservation of molecular mechanisms of disease, which can be built upon in further studies using flies to prove causal relationships between the defects described here and neurodegeneration.The identification of specific neurons involved in olfactory dysfunction opens up potential avenues for diagnostic and therapeutic interventions.Weaknesses:The causal relationship between early olfactory dysfunction and later motor symptoms in PD remains unclear. It is also uncertain whether this early defect contributes to neurodegeneration or is simply a reflection of the sensitivity of olfactory neurons to cellular impairments. The study does not investigate whether the observed early olfactory impairment in flies leads to later DAN deficits. Additionally, the single-cell RNA sequencing analysis reveals several affected neuronal populations that are not further explored. The main weakness of the paper is the lack of conclusive evidence linking early olfactory dysfunction to later disease progression.

We agree that this is an interesting avenue to pursue and as indicated above in Figure 6 and in the reworked manuscript, we have now included data that strengthens the connection between early OPN defects and the later DAN dependent problems. Additional future work will be needed to elucidate the mechanisms of this cell-non autonomous effect.

The rationale behind the selection of specific mutants and neuronal populations for further analysis could be better qualified.

We have added further explanation in the reworked text.

**Recommendations for the authors:**

**Reviewer #1 (Recommendations For The Authors):**
Minor Comments:(1) Questions about the sequencing methods and analysis approaches. From reading the methods and main text, I was confused about aspects of the *Drosophila* single-cell profiling. Firstly, did the authors multiplex their fly samples?

No, we did not. Genotypes were separately prepared and sequenced, but they were all processed in parallel to avoid batch effects.

Secondly, it seems like there are two rounds of dataset integration performed, Harmony and Seurat's CCA-based method. This seems unorthodox. Could the authors comment on why they perform two integrations?

Thanks for pointing this out, this was a mistake in the methods section (copied from a much older version of the manuscript). In this manuscript, we only used harmony for dataset integration and removed the methods on Seurat-CCA.

Finally, for all dataset integrations please state in the main text how datasets were integrated (by age, genotype, etc).

Datasets were integrated by sample id, corresponding to individual libraries.

(2) The authors focus on OPNs with a really nice set of experiments. I noticed however that Kenyon cells were also dysregulated. What about Olfactory sensory neurons? Could the authors provide comments on this?

Olfactory sensory neurons are located in the antennae of the fly brain and were not captured by our analysis. However, the GH146-Gal4-specific rescue experiments indicate these sensory neurons are likely not severely functionally impaired. Kenyon cells are an interesting affected cell type to look at in future experiments, as they are directly connected to DANs.

(3) There are several citations of Jenett et al 2012 that seem wrong (related to single-cell datasets).

We are sorry for this and have corrected this in the text.

**Reviewer #2 (Recommendations For The Authors)**:(1) In the key resources table, a line called CG5010k.o. (chchd2k.o.) was mentioned, but was not used in the paper. The authors should remove it.

Sorry, this was from a previous older version of the manuscript. We fixed this.

(2) Why did the authors use human CDS for LRRK2, Rab39B, and PINK1, but fly CDS for Aux and Synj1? Is it based on the conservation of amino acid residues? Although the authors cited a review (Kalia & Lang, 2015) to justify the selection of the mutations, for the interest of a broad audience, it is recommended that the authors expand their introduction for the rationale of their selection, including the pathogenicity of each selected mutation, original human genetics evidence, conservation between fly and human.

(a) We used *Drosophila* cDNA for rescue experiments with aux and synj since knockin of the human homologues at the locus of these genes did not rescue its loss-offunction (lethality).

(b) We expanded the introduction to provide further explanation on the selection of our mutants we analyzed in this work. We picked these particular mutants, as they were the first we created in the context of a much larger collection of “PD flies” (see also [42]). We have made adaptations to the text to tone down the statement on the broad selection of mutants.

(3) Supplemental Figure 1a, is mRNA level normalized to an internal control? If not, it is not appropriate to compare the results directly from two primer sets, since each primer set may have different amplification efficiency.

We are sorry for the lack of information. Indeed, mRNA levels were determined using the Δ-Δ-CT method, where Ct values were first normalized to the housekeeping gene Rp49, and next expressed as a percent of endogenous *Drosophila* gene expression. We expanded the methods section and now also enlist the primers for Rp49 along with the other qPCR primers in Supplemental File 1.

(4) For Figure 2, it may be helpful to have a supplemental table or figure showcasing the clusters with significant changes (based on cell number-adjusted DEGs) for each model, i.e., what are those black cell clusters in Figure 2? "Thus, cellular identity and cellular composition are preserved in young PD fly models." In Figure S2A, the authors only show cell composition percentages for 3 cell clusters, are the bars 95% standard error?

The error bars in Supplemental Figure 2a represent the 95 % CI. We have included a new supplemental table with the number of cells per cell cluster for each mutant (Supplemental File 3).

What about the remaining 183 cell clusters? Are there any KI-model cell clusters that are statistically different than controls? What about the annotated cell types (e.g., the 81 with cell identities)? Please consider at least providing or pointing to a table to state how many have significant differences, or if there are truly none.

As mentioned above, we have included a new supplemental table with the number of cells per cell cluster for each mutant (Supplemental File 3).

(5) What are the rows in the sunburst plot in Figure 3a? Please be more descriptive in the figure legend or label the figure.

We have expanded on this in the figure legend and now also include a summary of the SynGO analysis in Supplemental File 7. In Figure 3a, a summary sunburst plot is presented, reflecting the GO terms (inner rings, indicated in a) with their subdivided levels (the complete list is provided in Supplemental File 7). In Figure 3a’ and a” the DEG data acquired from the different datasets (human vs fly) are applied to the sunburst plot where rings are color-coded according to enrichment Q-value.

(6) In Table S4, which clusters (in the table) have normalized residuals that are outside of the 95% confidence interval of the regression model displayed in Figure S2e? They use this analysis to adjust for cell number bias and point out the "most significant cell clusters" affected in each model. This may be helpful for readers who want to grab a full list of responsive clusters.

We have included this information in Supplemental File 5 (Tab “Cell types outside of CIs”) in the supplemental data of the manuscript.

(7) The human samples used all have different LRRK2 variants: for the crossspecies comparisons, do Lrrk flies have greater similarity to the human PD cases compared to the other fly models?

No, comparing the vulnerable gene signatures from each of the fly mutants to the DEGs from the human samples does not show any greater similarity between the LRRK mutants compared to the other mutants.

**Reviewer #3 (Recommendations For The Authors):**
Clarifications required:Some of the mutations used are not common PD-associated genes, the authors should explain the rationale behind using these particular mutants, and not using well-established fly models of PD (like for example GBA flies) or SNCA overexpression.

We opted to use knock-ins of mutations that are causal to Parkinsonism. Given flies do not express an alpha-synuclein homologue we were not able to add this ‘as such’ to our collection. Future work can indeed also include expression models or risk factor models (like GBA). As also requested by another reviewer, we did add further rationale and explanation to the genes we chose to analyze in this work.

Why starvation rather than lifespan for PD models? For the lifespan data shown there are no error bars, if the stats test is a log-rank or Cox proportional hazards (usually used in survival analysis, this should be stated), it would also be good to have the survival plots for all the survival during starvation, not just PINK1.

While starvation assays can provide valuable insights into acute metabolic and physiological stress responses, we acknowledge that lifespan is a critical parameter and would provide a more comprehensive understanding of the PD models in our study. Based on this consideration and the reviewer’s feedback we have removed the starvation data from the manuscript. Unfortunately, we did not perform lifespan experiments, which is why these data were not included in the manuscript. However, based on our observations (though not detailed analysis), all genotypes tested—except for the PINK1 mutants—appeared to have a normal lifespan. For PINK1 mutants, most flies died by 25 days of age. Therefore, we conducted our assays using 15-day-old PINK1 mutant flies.

Do the fly models used have different lifespans, and how close to death was the SING assay performed? Different mutations show different effects, most phenotypes are really mild (hRab39BG192R has no phenotype), and PINK1 has the strongest, are these simply reflections of how strong the model is?

The ages of flies we analyzed are indicated in the legend. As mentioned before, all but PINK1 mutants- had a normal life span: i.e. we did not detect abnormal low number of flies or premature death at 50 days of age, except for the PINK1 mutants tested in this manuscript where most flies died by 25 days of age. Therefore, we conducted our assays using 15-day-old PINK1 mutant flies.

Rab39G192R has no phenotype in the tests presented, suggesting no degeneration, why use RabG192R for scRNA seq? Seems an odd choice, the authors should explain.

Single-cell sequencing was initiated before the full phenotypic characterization of all mutants was completed. Although basic characterization of the Rab39^G192R^ mutant PD flies revealed either no significant phenotypes or only mild effects in the assays performed (Figure 1), the sequencing data provided additional insights into potential cellular and molecular alterations. Furthermore, all PD-mutant knock-ins, including Rab39^G192R^ mutant PD flies, show dysfunctional synaptic terminals of their OPN neurons as they had significantly weaker Ca^2+^-responses, even though their synaptic area was increased (Figure 4 g-h). Furthermore, all mutants also had olfactory behavior defects (Figure 5 a).

When the authors state that “For example, in the NBM, an area associated with PD (Arendt et al., 1983), 20% of the DEG that has an orthologous gene in the fly are also found among the most deregulated genes across PD fly models" a test should be performed to confirm this is a significant overlap (such as a hypergeometric test).

We have performed this test, of the 2486 significantly differential human genes, 1149 have a fly orthologue, and of these, 28.46 % overlap with the deregulated fly genes (5 % top and bottom gene as shown in Supplemental Table 7). Performing a hypergeometric test confirms that this overlap is significant, with a p-value of 9.06e^76^. We have included this in the text.

The authors speak of deregulation when speaking of the overlap between human and fly DE genes, but do the over-expressed genes in flies overlap with overexpressed genes in humans, or is the direction of transcription deregulation not concordant? If it is mostly not concordant, can the authors please comment as to why they might think that is the case?

In our fly experiments, we identified DEG in affected cell types and then defined common DEG by looking at the average change across the fly mutants. Genes that show a consistent change (all or mostly up, or all or mostly down) in the different mutants will end at the top of our list while genes that are up in some mutants and downregulated in others will average out and not end up in our commonly deregulated gene list. For comparison to the human data, we only looked for the presence of the human homologue, but did not assess if the change occurred in the same direction. More work will be needed to define the most relevant changes, but in a mini-screen we did select a number of DEG present in fly and human datasets from different functional categories and tested if they genetically interact with our PD mutants. As shown in Author response image 1, we find that modulating proteostasis pathway-encoding genes rescue the olfactory preference defect across many PD mutants.

Can the authors explain why only the NMB region was used for comparison with the fly data?

We used the NMB because this region has the highest number of cholinergic neurons to compare the deregulation in those neurons to the deregulation in the cholinergic OPN of mutant PD flies.

In Figure 4, can the genotypes please be stated in full and why is the hPINK1 fly giving no detectable signal?

Despite several attempts, we failed to knock-in wild type hPink1 in the fly pink1 locus. Therefore, the hPink1 control used throughout the manuscript was the nSybGal4>UAS-hPink1 in Pink1 knock-out background, except for Figure 4. Particularly, for experiments in this figure, we could not use UAS-hPink1 with nSyb-Gal4, since we needed OPN-specific expression of Gal4 to drive UAS-GCamP expression.

Therefore, this was labeled as “not determined” (“n.d.”), as indicated in the figure and the legend. We explained this better in the methods section, added a remark in the new manuscript and expanded the legend of Figure 4.

The paper states that" These findings imply that factors affecting the function of cholinergic neurons might, by the absence of insufficient innervation, lead to DAN problems and degeneration, warranting further exploration of the underlying molecular mechanisms", this should be less strong, the paper never looks at DAN, only at OPN neurons. Fly neurons are mostly cholinergic, and human neurons are mostly glutamatergic, so jumping from one system to the other might not be as straightforward, the authors should comment on this.

We now included a new exciting experiment where we assessed DAN function in aged PD mutants where the wildtype gene was expressed in OPN using GH146-Gal4. We find this manipulation rescued DAN defects (measured by SING) in older flies. We further corroborated our observation by “replacing” cholinergic innervation with nicotine feeding in PD mutants. Also, this rescues the SING defect as well as the defects in neuronal activity in PAM DAN (based on live synaptic calcium imaging). Finally, we also show that incubating LRRK2^G2019S^ mutant human induced dopaminergic neurons with nicotine is sufficient to rescue functional defects in these neurons (measured using calcium imaging). We included this data in the new manuscript and show them also in Figure 6 above (new Figure 6 in the revised manuscript).

Experiments that would improve the manuscript:Does rescue of OPN function also rescue later progressive symptoms (geotaxis response)?

It does, as indicated in the previous point and shown in Figure 6.

Do the fly PD models used show DAN degeneration? This could be assessed by stains with anti-TH stains.

We quantified DAN cell bodies using anti-TH, but see very little or no loss. There is, however, loss of synaptic innervation of the PAM onto the mushroom bodies. We included the data in a new Figure 6 (see also Figure 6). Furthermore, we have quantified this across the genetic space of familial Parkinsonism in [42]. Note that this phenotype is also rescued by expressing wildtype CDS in their OPN using GH146-Gal4.

Minor issues:The final sentence on page 5 is repetitive with the introduction.

Indeed, we removed the redundant sentence.

First line of the new section on page 6, the authors probably mean cholinergic olfactory projection neurons, not just cholinergic neurons.

Yes, and corrected.

At the top of page 7 the authors state: "Additionally, we also found enrichment of genes involved in RNA regulation and mitochondrial function that are also important for the functioning of synaptic terminals", where is the data showing this? The authors should point to the supplemental file showing this.

We now included a reference to Supplemental File 7 that includes a summary of those data. Additionally, we also included references to back this claim.

Just before the discussion, Rab39BG193R should be Rab39BG192R.

Sorry for this, it is now corrected.

Stating "fifth row" in Fig 5c and d is confusing, can the figure be labelled more clearly?

We modified the figure (including extra marks and colors) and expanded the legend and the main text to differentiate better between expression of the rescues in OPN versus T1 neurons revealing that only expression in OPN neurons rescues the olfactory defects while expression in T1 neurons does not.

In the methods, the authors describe clustering done both in Scanpy and Seurant, why were both run? Which clustering was used for further analysis?

We only used Scanpy with Harmony and removed the methods on Seurat-CCA. Thanks for pointing this out, this was a mistake in the methods section (copied from a previous version of the manuscript).